# *Ask Me in English Instead*: Cross-Lingual Evaluation of Large Language Models for Healthcare Queries

Submission Id: 1949*

## ABSTRACT

Large language models (LLMs) are transforming the ways the general public accesses and consumes information. Their influence is particularly pronounced in pivotal sectors like healthcare, where lay individuals are increasingly appropriating LLMs as conversational agents for everyday queries. While LLMs demonstrate impressive language understanding and generation proficiencies, concerns regarding their safety remain paramount in these high-stake domains. Moreover, the development of LLMs is disproportionately focused on English. It remains unclear how these LLMs perform in the context of non-English languages, a gap that is critical for ensuring equity in the real-world use of these systems. This paper provides a framework to investigate the effectiveness of LLMs as multilingual dialogue systems for healthcare queries. Our empirically-derived framework XLINGEVAL focuses on three fundamental criteria for evaluating LLM responses to naturalistic human-authored health-related questions: correctness, consistency, and verifiability. Through extensive experiments on four major global languages, including English, Spanish, Chinese, and Hindi, spanning three expert-annotated large health Q&A datasets, and through an amalgamation of algorithmic and human-evaluation strategies, we found a pronounced disparity in LLM responses across these languages, indicating a need for enhanced cross-lingual capabilities. We further propose XLINGHEALTH, a cross-lingual benchmark for examining the multilingual capabilities of LLMs in the healthcare context. Our findings underscore the pressing need to bolster the cross-lingual capacities of these models, and to provide an equitable information ecosystem accessible to all.

## CCS CONCEPTS

• **Applied computing → Health care information systems**; **Health informatics**; • **Computing methodologies → Natural language generation**; **Language resources**.

## KEYWORDS

large language model, natural language processing, cross-lingual evaluation, language disparity

**ACM Reference Format:**
Anonymous Author(s). 2024. *Ask Me in English Instead*: Cross-Lingual Evaluation of Large Language Models for Healthcare Queries. In *Proceedings of the ACM Web Conference 2024 (WWW'24)*. ACM, New York, NY, USA, 18 pages. https://doi.org/XXXXXXX.XXXXXXX

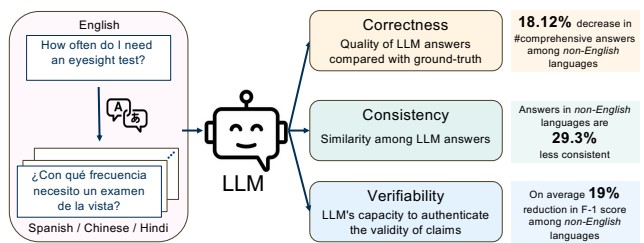

**Figure 1: We present XLINGEVAL, a comprehensive framework for assessing cross-lingual behaviors of LLMs for high risk domains such as healthcare. We present XLINGHEALTH, a cross-lingual benchmark for healthcare queries.**

## 1 INTRODUCTION

Large language models (LLMs) have gained popularity due to their ability to understand human language and deliver exceptional performances in various tasks [1–3]. While LLMs have been used by experts for downstream generative tasks [4, 5], their recent adoption as dialogue systems has made them accessible to the general public, especially with models like GPT-3.5 [6], GPT-4 [7], and Bard [8] becoming widely available [9]. This expanded availability to LLMs is expected to enhance access to education, healthcare, and digital literacy [10, 11]. Especially in healthcare, LLMs exhibit significant potential to simplify complex medical information into digestible summaries, answer queries, support clinical decision-making, and enhance health literacy among the general population [12, 13]. However, their adoption in healthcare domain brings two significant challenges: ensuring safety and addressing language disparity.

Safety concerns associated with individuals, especially those without specialized expertise who heavily depend on LLMs in critical domains like healthcare, require significant attention. In such fields, where incorrect or incomplete information can have life-threatening consequences, overreliance on or misinterpretation of the information provided by these models represents a substantial and pressing challenge. However, past work has predominantly focused on evaluating the knowledge capabilities of LLMs, leaving a gap in understanding the characteristics pertaining to the quality of interactions between humans and LLMs. Consequently, it is vital to assess the safety of LLM behaviors, including their ability to provide consistent, correct, and comprehensive answers to healthcare queries and authenticate claims accurately.

Furthermore, in the domain of model training and evaluation, there exists a notable *language disparity* [14], a phenomenon where a significant emphasis is centered around the English language [7, 15]. Such an inclination can compromise the principle of equitable

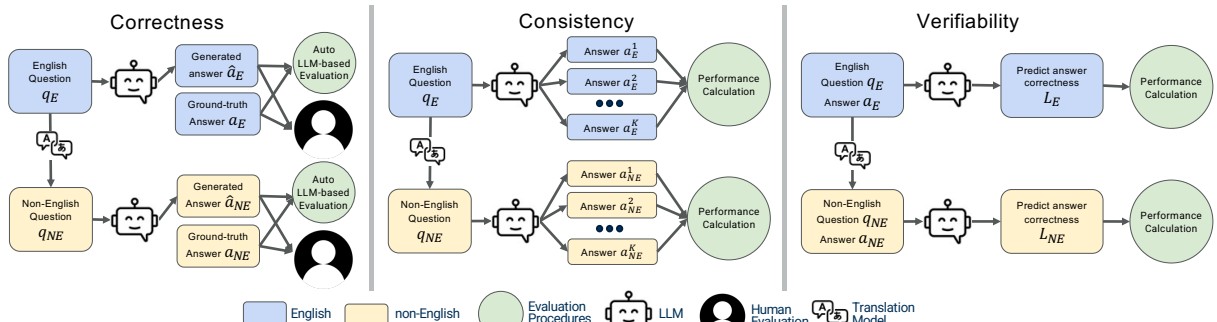

**Figure 2: Evaluation pipelines for correctness, consistency, and verifiability criteria in the XLingEval framework.**

access, especially given that more than 82% of the global population does not speak English as their primary or secondary language [16], thus impacting billions of non-native English speakers worldwide. In light of the paramount importance of ensuring equal access to health-related information, it becomes evident that solely focusing on LLMs' safety evaluations in English is inadequate. Instead, a comprehensive, multilingual evaluation approach is needed to effectively address language disparity.

In response to these challenges, we propose **XLingEval** a comprehensive cross-lingual framework to assess the behavior of LLMs, especially in high-risk domains such as healthcare. Our framework emphasizes the importance of **equity across languages** and **generalizability across models**, guided by our proposed evaluation metrics for LLM evaluations. We specifically propose three *criteria* for conversational language models:

- **Correctness**: The model's responses should exhibit factual correctness and comprehensively address the query.
- **Consistency**: The model should produce consistent responses to identical queries, reflecting high similarity in lexical, semantic, and topic aspects.
- **Verifiability**: The model should be capable to authenticate accurate claims and clearly distinguish between correct and erroneous responses to a query.

The **cross-lingual equity** dimension within our framework emphasizes on evaluating the cross-lingual capabilities of LLMs. We propose a comparative evaluation of the aforementioned criteria across the four most widely spoken languages in the world — *English*, *Hindi*, *Chinese*, and *Spanish* [17]. Additionally, the **generalizability** aspect of our framework centers on conducting cross-lingual evaluations on other LLMs, such as MedAlpaca, a specialized language model fine-tuned on medical documents) [18] and adapting the proposed framework for other domains.

Our experiments reveal a discernible disparity across languages in all three evaluation metrics. Regarding **correctness** (Section 3), we observe an average decrease of 18.12% in the number of 'more comprehensive and appropriate answers' produced by GPT-3.5 when responding to queries in *Non-English* languages as compared to *English* across the three datasets. However, for *Non-English* languages, GPT-3.5 is 5.82 times more likely to produce incorrect responses than in *English*. Regarding **consistency** (Section 4), GPT-3.5 tends to generate more consistent responses on English compared to non-English languages. We observe a maximum performance decrease of 9.1% in Spanish, 28.3% in Chinese, and 50.5% in

Hindi when compared to English. All language pairs, except English-Spanish, exhibit statistically significant differences in performance, demonstrating the existence of language disparity. Regarding **verifiability** (Section 5), English and Spanish demonstrate comparable performances, whereas the performances for Chinese and Hindi are notably lower. In the most extreme case, Chinese and Hindi exhibit decreases of 14.6% and 23.4% on Macro F1, respectively.

Our research carries significant real-world implications on multiple fronts. The evaluation framework proposed in our work possesses practical utility for policymakers, practitioners, and healthcare professionals for evaluating large language models and comparing their relative performance. Through our examination of LLMs' capabilities in major languages, we aspire to acquire a comprehensive understanding of their global effectiveness, which stands to influence a vast and linguistically diverse global population, impacting both linguistic accessibility and information reliability. Furthermore, our framework exhibits versatility and adaptability beyond healthcare, extending its applicability to other domains.

Our contributions are summarized as follows:

- **Novel Framework**. We propose XLingEval, a comprehensive evaluation framework for LLMs in the healthcare domain that focuses on three fundamental criteria: *correctness, verifiability,* and *consistency*. Our framework features the gaps in *equity* in LLM development across multiple languages, and demonstrates *generalizability* in this evaluation across different LLMs.
- **Novel Medical Benchmark.** We propose XLingHealth, a Cross-Lingual Healthcare benchmark for clinical health inquiry that features the top four most spoken languages in the world.
- **Extensive Multilingual Evaluation.** We performed comprehensive evaluation on the four most spoken languages, and found significant *language disparity* across these languages.

Our code and data are available on Anonymous GitHub[1]. We will make all the code, data, and tools publicly available upon the acceptance of this work.

## 2 THE XLINGHEALTH BENCHMARK

Our proposed XLingHealth is a novel cross-lingual healthcare benchmark for clinical health inquiry. It is based on three prominent healthcare datasets consisting of question-and-answer pairs curated by medical expert. A brief introduction is provided below, with additional statistical details available in the Appendix Table A1.

---

[1]https://anonymous.4open.science/r/llm

- **HealthQA** [19]. This dataset is constructed using specialized healthcare articles on the popular health service website Patient [20]. The questions are created by a diverse range of annotators from the health topics sections, and the answers are excerpts from the original articles. We use the dev set comprising of 1,134 questions for our experiments, where each question has one correct answer and 9 incorrect ones.
- **LiveQA** [21]. This dataset contains 246 question-answer pairs constructed using frequently asked questions (FAQs) from trusted platforms associated with U.S. National Institutes of Health (NIH).
- **MedicationQA** [22]. This dataset contains 690 examples. The questions, primarily address drug-related concerns, are extracted from anonymized consumer queries submitted to MedlinePlus [23]. The answers are sourced from medical references such as MedlinePlus and DailyMed [24].

The selection of these datasets aligns with general public health queries. The questions closely resemble those typically asked by the general public, ensuring their relevance to a broader audience that may lack specialized medical knowledge. The answers are provided by medical professionals, enhancing the credibility and reliability of the data sources. However, it is important to note that these datasets are originally in English. Given the scarcity of multilingual health and medical question-answering datasets, we create a novel multilingual benchmark by translating these datasets into Hindi, Chinese, and Spanish. To ensure the dataset quality, we performed a comprehensive human evaluation (further details in Appendix A).

Next, we turn our attention to our proposed **XLingEval**, a comprehensive evaluation framework for LLMs in the healthcare domain in the following sections.

## 3 CORRECTNESS

The first fundamental criterion of XLINGEVAL is correctness, which pertains to the accuracy, comprehensiveness, and contextual appropriateness of LLMs' responses in healthcare inquiries. Ensuring correctness is essential due to the substantial implications associated with inaccuracies or errors in responses [22, 25–27]. To evaluate the correctness criterion in XLINGEVAL, we conducted experiments to compare LLMs' responses to expert-curated ground-truth answers across the three healthcare datasets:

For the evaluation criteria, we merged and modified the categories from the past work [27] to assess two key relationships between the answers: 1) Contradiction and 2) Comprehensiveness & Appropriateness. Contradiction refers to the incorrect or contrasting information provided in the LLM answer compared to the Ground Truth answer. Comprehensiveness refers to the details provided in the answer and whether it covers the points/topics expected from the answer. Appropriateness gauges how well the answer aligns with the context provided in the question. These relationships are represented by four classification labels as shown in Table 1. We present the rationale for the selection of axis labels and elucidate how these labels effectively depict the respective axes in Appendix B. Finally, the evaluation setup for the correctness criterion consists of two components: 1) Automated Evaluation, for large-scale and statistically significant comparisons between the LLM answers and ground-truth, and 2) Human Evaluation, which serves as validation for the automated evaluation.

### 3.1 Automated Evaluation

The automated evaluation for correctness encompassed two phases, as depicted in the flowchart in Figure 2. In *Phase-1*, the LLM (GPT-3.5) was prompted with questions from each dataset, yielding an LLM answer for each question. In *Phase-2*, we conducted a comparative analysis between the LLM answer and the ground-truth answer from the dataset. Specifically, we prompted the LLM with the question, ground-truth answer, and the LLM answer using Chain-of-Thought (CoT) prompting [28]. We asked the LLM to assign one of the four labels in Table 1. In the *Phase-2* prompt, the initial instruction directed the LLM to assess whether the LLM answer contradicted or found similar with the ground-truth. If found similar, subsequent instructions prompted the LLM to compare the comprehensiveness and appropriateness of the answers. The prompts are detailed in Appendix Table A3.

**Findings for comprehensiveness and appropriateness:** Table 1 presents the results for the automated comparative evaluation. Across all datasets, we observed a drastic decrease in the number of examples where GPT-3.5 provides more comprehensive and appropriate answers compared to the ground-truth answers. For HealthQA, we observed a relative decrease in the number of GPT-3.5 answers providing more comprehensive and appropriate information by 38.62% for *Hindi* answers, 11.90% for *Chinese* answers and 10.76% for *Spanish* answers as compared to that of answers in *English*. We observed a similar trend for LiveQA having a relative decrease of 34.15%, 5.69%, and 5.28% for *Hindi*, *Chinese*, and *Spanish* respectively. For MedicationQA, we observed a relative decrease of 30.58%, 15.8%, and 10.29% for answers where GPT-3.5 produced more comprehensive and appropriate answers in *Hindi*, *Chinese*, and *Spanish* respectively.

**Findings for contradiction:** Meanwhile, the number of GPT-3.5 answers in *Hindi*, *Chinese*, and *Spanish* with contradictory information increased compared to the ground-truth answers, relative to the answers GPT-3.5 provided in *English*. While GPT-3.5 produced 3 contradictory answers in *English* for the HealthQA dataset, it produced 47 (15.67 times) contradictory answers for *Hindi*, 14 (4.67 times) for *Chinese*, and 5 (1.67 times) for *Spanish*. For Live QA we observed GPT-3.5 producing 4.33 times more contradictory answers in *Hindi* as compared to *English*. Finally, for Medication QA dataset, we observed a huge increase in the number of contradictory answers with GPT-3.5 producing 51 (10.2 times) in *Hindi*, 48 (9.6 times) in *Chinese*, and 23 (4.6 times) in *Spanish*. Finally, we performed the same set of analyses for MedAlpaca and observed a similar disparity between the performance for English and non-English languages. In contrast to the GPT-3.5 results, we observed a drastic increases in the number of answers procued by MedAlpaca which were neither contradictory no similar to the Ground Truth. This can be attirbuted to the incapibility of MedAlpaca to produce multilingual texts. Detailed analysis on MedAlpaca results in provided in Appendix D.3.

Overall, we observed language disparity across all four evaluation labels from the automated evaluation, with *Hindi* showing the most prominent discrepancy, followed by *Chinese* and *Spanish*.

### 3.2 Human Evaluation

In addition to the automated evaluation, we also conducted an IRB-approved human evaluation as a validation measure for the

**Table 1: Automated correctness evaluation in four languages: English (en), Spanish (es), Chinese (zh), and Hindi (hi) for GPT-3.5. Each number represents the number of answers assigned to the respective label in the dataset.**

| Information Comparison (LLM Answer vs ground-truth Answer) | HealthQA | | | | LiveQA | | | | MedicationQA | | | |
|---|---|---|---|---|---|---|---|---|---|---|---|---|
| | en | es | zh | hi | en | es | zh | hi | en | es | zh | hi |
| More comprehensive and appropriate | 1013 | 891 | 878 | 575 | 226 | 213 | 212 | 142 | 618 | 547 | 509 | 407 |
| Less comprehensive and appropriate | 98 | 175 | 185 | 402 | 3 | 12 | 16 | 59 | 18 | 50 | 41 | 125 |
| Neither contradictory nor similar | 20 | 63 | 57 | 110 | 14 | 20 | 14 | 32 | 49 | 70 | 92 | 107 |
| Contradictory | 3 | 5 | 14 | 47 | 3 | 1 | 4 | 13 | 5 | 23 | 48 | 51 |

large-scale automated evaluation. The human evaluation involved constructing an annotation dataset generated by randomly selecting 10% of examples from a stratified pool drawn from the three datasets. Stratification was determined by the distribution of examples across the four labels in Table 1 assigned by GPT-3.5. In total, we assembled a corpus of 103 such instances for each language. Each instance within this annotation dataset comprised a quadruple, consisting of a question, an expert-curated answer, a response generated by the LLM, and a reasoning generated from GPT-3.5 (during the *Phase-2* prompting) that elucidated the justification behind the classification label ascribed to the given example. The annotators were required to answer a yes/no based question on whether they agreed with the reasoning and classification label provided by the LLM. Additionally, in cases where the annotator did not agree with the reasoning, we asked them to provide the reasoning for selecting the 'no' option along with reporting the correct relationship between the two answers (more details in Appendix C). We assigned the majority label to each instance based on the annotations.

We leveraged various channels for hiring medical experts for this task, including crowd-sourcing platforms such as Prolific, social media platforms like Reddit, LinkedIn, and traditional recruitment methods such as mailing lists. To facilitate the annotation process, we developed a novel web application as detailed in Appendix C. We divided the examples for each language into two batches (batch-1 and batch-2), and each subset was annotated by three annotators. This division was necessitated by the exhaustive nature of our annotation task. Assessing the undivided set could have required over 6 hours, potentially leading to high dropout rates and compromised response quality due to annotator fatigue.

In the case of the *English* examples, our analysis revealed a notable average correlation of 94.20% between the labels ascribed by GPT-3.5 and the majority labels from human annotators. Moreover, on average, all three human annotators unanimously agreed with GPT-3.5's labeling in 74.74% of the instances. For *Spanish*, we observed the average correlation to be 95.14%. Despite employing a thorough methodology in our search for medical experts as annotators, we encountered difficulties in securing the requisite number of three annotators for each batch, specifically in the case of the *Chinese* and *Hindi* language, for which we could only enlist two annotators, and one annotator respectively. Detailed results of the human evaluation are provided in Appendix C. We observed an average correlation of 77.61% for *Chinese*, and 84.47% for *Hindi*. The human evaluation served as a corroborative measure to validate the credibility and reliability of our automated evaluation approach.

## 4 CONSISTENCY

The second critical criterion in XLɪɴɢEᴠᴀʟ is consistency. Assessing the consistency of LLM's responses has become crucial and pertinent in areas that require precision and reliability, such as healthcare. Inconsistent medical guidance provided by these models can mislead patients, diminishing the credibility of LLMs, and impacting the well-being of individuals. To address this challenge, the consistency criterion protocol gauges the coherence of LLM-generated responses. To achieve this, we varied the "temperature" parameter $\tau$ of language models to control the randomness of the generated text. As shown in Figure 2, for each question, we prompt the LLM $K = 10$ times using both the English version ($q_E$) and non-English version ($q_{NE}$) of the same question. Then, for each question, we measure the similarity among answers $\{a_E^k\}_{k=1}^K$ and $\{a_{NE}^k\}_{k=1}^K$ respectively according to the metrics described below. The responses are evaluated across multiple dimensions, ranging from surface-level, semantic-level, and topic-level.

### 4.1 Metrics

*4.1.1 Surface-level Consistency.* Surface-level consistency gauges the resemblance between two pieces of text based on their superficial attributes, such as lexical features, word choices and response lengths, disregarding the deeper contextual or semantic meaning.

**N-gram Similarity** ($\text{sim}_{\text{n-gram}}$) [29, 30] is the Jaccard similarity between the set of n-grams present in the two documents:

$$\text{sim}_{\text{n-gram}}(s_1, s_2) = \frac{|\text{n-grams}(s_1) \cap \text{n-grams}(s_2)|}{|\text{n-grams}(s_1) \cup \text{n-grams}(s_2)|}, \quad (1)$$

where $s_1, s_2$ are two generated answers for comparison, and n-grams($s_1$) indicates the set of n-grams in $s_1$. Here, we consider unigram and bigram similarity, i.e., $n = 1, 2$.

**Length of Response** is defined as the number of words in the answer, excluding punctuation marks and spaces. For cross-lingual evaluation, we translate LLM-generated non-English answers back to English using the procedures in Appendix A.

*4.1.2 Semantic-level Consistency.* Semantic-level consistency [31, 32] measures the semantic association between two answers. This type of assessment requires a deep understanding of subject matters described in different responses. For example, words like "obesity" and "BMI" have different meanings but often exhibit strong semantic associations due to their frequent co-occurrence in discussions about weight control. To assess semantic similarity, we leveraged two metrics based on contextualized word embeddings, known for capturing distant dependencies [33–35] and their strong correlation with human judgments [36, 37]. Specifically:

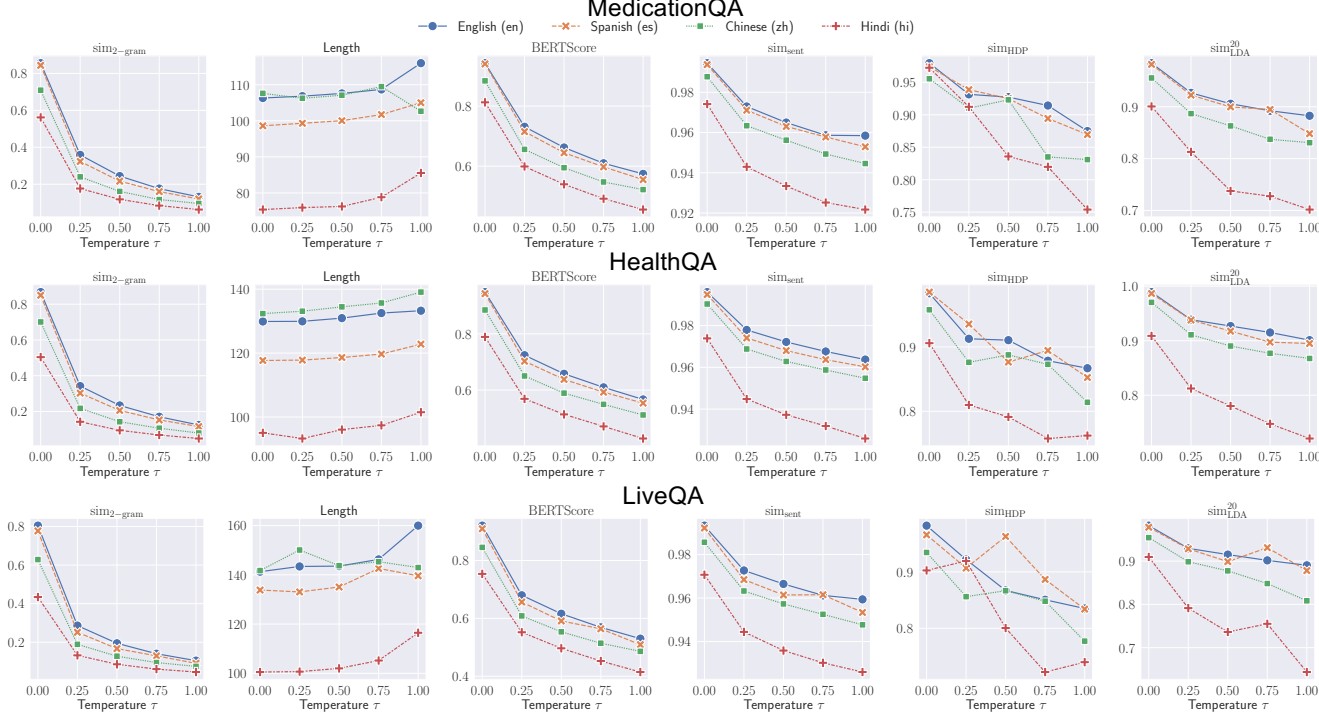

**Figure 3: Results of consistency metrics on the three datasets. Each row represents the results of a particular dataset, and each column indicates a distinct metric.**

**BERTScore** [38] leverages contextualized embeddings to capture a token's specific usage in a sentence and potentially incorporates sequence information.

**Sentence Embedding Similarity** ($sim_{sent}$) [39] is the cosine similarity between the sentence embeddings of two answers:

$$sim_{sent}\left(\mathbf{s}_i, \mathbf{s}_j\right) = \frac{\mathbf{s}_i \cdot \mathbf{s}_j}{\|\mathbf{s}_i\|\|\mathbf{s}_j\|}, \qquad (2)$$

where $\mathbf{s}_i$ is the embedding of the $i$-th response. We leveraged Sentence-BERT [39] to encode each response into a 768-dimensional representation with bert-base-uncased as the base model.

*4.1.3 Topic Consistency.* Topic similarity measures whether two answers discuss similar topics from a macro perspective [40–43]. Quantitative assessment of topic similarity through human evaluation can be challenging as it is hard to assign precise scores to generated answers that exhibit varying levels of similarity due to different temperature settings ($\tau$). To address this challenge, we employed two topic modeling techniques to quantify topic similarity:

**Latent Dirichlet Process (LDA)** [44] represents documents as mixtures of topics and infers the underlying topic distribution of each response. When using a topic number of $n$, the topic similarity between two answers $s_i, s_j$ is defined as:

$$sim_{LDA}^n(s_i, s_j) = \frac{\mathbf{t}^n(s_i) \cdot \mathbf{t}^n(s_j)}{\|\mathbf{t}^n(s_i)\|\|\mathbf{t}^n(s_j)\|}, \qquad (3)$$

where $\mathbf{t}^n(s_i) \in \mathbb{R}^n$ is the topic distribution of $s_i$ when the number of topics is set to $n$. Note that LDA requires a predefined number of topics and may generate closely aligned or duplicated topics when the number is large.

**Hierarchical Dirichlet Process (HDP)** [45] is a non-parametric Bayesian technique which automatically infers the optimal number of topics based on the complexity and volume of the data. Empirically, we fitted a topic model to the complete set of LLM-generated answers on each dataset for a single language, and subsequently derived a topic distribution for each individual answer.

## 4.2 Results

*4.2.1 Numerical Results.* Figure 3 illustrates the consistency of GPT-3.5's outputs based on different evaluation metrics. In terms of $sim_{2\text{-}gram}$, BERTScore, and $sim_{sent}$, GPT-3.5 exhibited higher consistency in its answers in *English* compared to other languages. For BERTScore, GPT-3.5 achieved 0.9206 / 0.6160 / 0.5299 for $\tau = 0.0$ / 0.5 / 1.0, whereas its performances in *Chinese* dropped to 0.8454 / 0.5536 / 0.4860 for the same $\tau$ values. The performance disparity between GPT-3.5's performance in *English* and *Spanish* is relatively narrow compared to the other languages. For BERTScore, GPT-3.5 demonstrates performances of 0.9097 / 0.5910 / 0.5092 under the three temperatures for *Spanish*, which are comparable to its performances in *English*. It is noteworthy that GPT-3.5 demonstrated relatively high semantic-level consistency in terms of $sim_{sent}$. On LiveQA (Table A9), the model yielded average scores of 0.9706, 0.9674, 0.9613 and 0.9415 across the four languages, with a modest maximum performance decrease of 3.0% compared with *English*. This high semantic consistency stood in stark contrast to its surface-level consistency, where GPT-3.5 manifested a maximum decrease of -50.7% on *Hindi* compared with *English* $sim_{2\text{-}gram}$. This suggested that, while GPT-3.5 can maintain semantic consistency even with escalating generative randomness, there are pronounced shifts in

**Table 2: Tukey's HSD test results on LiveQA ($\tau = 0.0$). We report the 95% confidence interval (95% CI) and the mean difference (MD). Asterisks (*) denotes the significance level. * / ** / *** stand for $p < 0.05 / 0.01 / 0.001$, respectively. Significant disparities were observed on all metrics across all language pairs, with the exception of English and Spanish.**

| $\tau = 0.0$ | | $\text{sim}_{\text{BERT}}$ | | | BERTScore | | | $\text{sim}_{\text{1gram}}$ | | |
|---|---|---|---|---|---|---|---|---|---|---|
| Language | | 95% CI | MD | p-adj | 95% CI | MD | p-adj | 95% CI | MD | p-adj |
| en | es | (-0.0037, 0.0025) | -0.0006 | 0.9535 | (-0.0286, 0.0164) | -0.0061 | 0.8962 | (-0.0442, 0.0234) | -0.0104 | 0.8576 |
| en | zh | (-0.0106, -0.0044) | -0.0075 | <0.001*** | (-0.0958, -0.0508) | -0.0733 | <0.001*** | (-0.1578, -0.0903) | -0.124 | <0.001*** |
| en | hi | (-0.0258, -0.0196) | -0.0227 | <0.001*** | (-0.1891, -0.1441) | -0.1666 | <0.001*** | (-0.3090, -0.2415) | -0.2752 | <0.001*** |
| es | zh | (-0.0100, -0.0038) | -0.0069 | <0.001*** | (-0.0897, -0.0447) | -0.0672 | <0.001*** | (-0.1474, -0.0799) | -0.1136 | <0.001*** |
| es | hi | (-0.0252, -0.0190) | -0.0221 | <0.001*** | (-0.1830, -0.1380) | -0.1605 | <0.001*** | (-0.2986, -0.2311) | -0.2648 | <0.001*** |
| zh | hi | (-0.0183, -0.0121) | -0.0152 | <0.001*** | (-0.1158, -0.0708) | -0.0933 | <0.001*** | (-0.1850, -0.1174) | -0.1512 | <0.001*** |

| $\tau = 0.0$ | | $\text{sim}_{\text{2grams}}$ | | | $\text{sim}_{\text{LDA}}^{20}$ | | | $\text{sim}_{\text{HDP}}$ | | |
|---|---|---|---|---|---|---|---|---|---|---|
| Language | | 95% CI | MD | p-adj | 95% CI | MD | p-adj | 95% CI | MD | p-adj |
| en | es | (-0.0616, 0.0265) | -0.0175 | 0.7349 | (-0.0210, 0.0155) | -0.0028 | 0.9800 | (-0.0401, 0.0065) | -0.0168 | 0.2478 |
| en | zh | (-0.2153, -0.1272) | -0.1712 | <0.001*** | (-0.0443, -0.0079) | -0.0261 | 0.0014** | (-0.0713, -0.0248) | -0.0480 | <0.001*** |
| en | hi | (-0.4142, -0.3262) | -0.3702 | <0.001*** | (-0.0923, -0.0559) | -0.0741 | <0.001*** | (-0.1068, -0.0603) | -0.0836 | <0.001*** |
| es | zh | (-0.1977, -0.1097) | -0.1537 | <0.001*** | (-0.0416, -0.0051) | -0.0234 | 0.0055** | (-0.0545, -0.0080) | -0.0312 | 0.0032** |
| es | hi | (-0.3967, -0.3086) | -0.3527 | <0.001*** | (-0.0896, -0.0532) | -0.0714 | <0.001*** | (-0.0900, -0.0435) | -0.0668 | <0.001 |
| zh | hi | (-0.2430, -0.1549) | -0.1989 | <0.001*** | (-0.0662, -0.0298) | -0.048 | <0.001*** | (-0.0588, -0.0122) | -0.0355 | <0.001*** |

**Table 3: Tukey's HSD test results on LiveQA with $\tau = 1.0$.**

| $\tau = 1.0$ | | $\text{sim}_{\text{BERT}}$ | | | BERTScore | | | $\text{sim}_{\text{1gram}}$ | | |
|---|---|---|---|---|---|---|---|---|---|---|
| Language | | 95% CI | MD | p-adj | 95% CI | MD | p-adj | 95% CI | MD | p-adj |
| en | es | (-0.0104, -0.0018) | -0.0061 | 0.0017** | (-0.0317, -0.0108) | -0.0212 | <0.001*** | (-0.0252, -0.0106) | -0.0179 | <0.001*** |
| en | zh | (-0.0161, -0.0075) | -0.0118 | <0.001*** | (-0.0542, -0.0332) | -0.0437 | <0.001*** | (-0.0523, -0.0376) | -0.0450 | <0.001*** |
| en | hi | (-0.0376, -0.0289) | -0.0332 | <0.001*** | (-0.1259, -0.1049) | -0.1154 | <0.001*** | (-0.0960, -0.0814) | -0.0887 | <0.001*** |
| es | zh | (-0.0100, -0.0014) | -0.0057 | 0.0037** | (-0.0329, -0.0120) | -0.0225 | <0.001*** | (-0.0344, -0.0198) | -0.0271 | <0.001*** |
| es | hi | (-0.0315, -0.0229) | -0.0272 | <0.001*** | (-0.1046, -0.0837) | -0.0942 | <0.001*** | (-0.0782, -0.0635) | -0.0708 | <0.001*** |
| zh | hi | (-0.0258, -0.0171) | -0.0214 | <0.001*** | (-0.0822, -0.0612) | -0.0717 | <0.001*** | (-0.0511, -0.0364) | -0.0438 | <0.001*** |

| $\tau = 1.0$ | | $\text{sim}_{\text{2grams}}$ | | | $\text{sim}_{\text{LDA}}^{20}$ | | | $\text{sim}_{\text{HDP}}$ | | |
|---|---|---|---|---|---|---|---|---|---|---|
| Language | | 95% CI | MD | p-adj | 95% CI | MD | p-adj | 95% CI | MD | p-adj |
| en | es | (-0.0189, -0.0088) | -0.0138 | <0.001*** | (-0.0430, 0.0192) | -0.0119 | 0.7579 | (-0.0343, 0.0336) | -0.0004 | 0.9909 |
| en | zh | (-0.0346, -0.0244) | -0.0295 | <0.001*** | (-0.1108, -0.0486) | -0.0797 | <0.001*** | (-0.0865, -0.0187) | -0.0526 | 0.0004** |
| en | hi | (-0.0643, -0.0541) | -0.0592 | <0.001*** | (-0.2785, -0.2164) | -0.2475 | <0.001*** | (-0.1250, -0.0571) | -0.091 | <0.001*** |
| es | zh | (-0.0207, -0.0106) | -0.0157 | <0.001*** | (-0.0989, -0.0367) | -0.0678 | <0.001*** | (-0.0862, -0.0183) | -0.0522 | <0.001*** |
| es | hi | (-0.0504, -0.0402) | -0.0453 | <0.001*** | (-0.2666, -0.2045) | -0.2356 | <0.001*** | (-0.1246, -0.0567) | -0.0906 | <0.001*** |
| zh | hi | (-0.0348, -0.0246) | -0.0297 | <0.001*** | (-0.1989, -0.1367) | -0.1678 | <0.001*** | (-0.0724, -0.0045) | -0.0384 | 0.0191* |

its lexical selections. In general, GPT-3.5 demonstrated the highest and lowest consistency on *English* and *Hindi*, respectively.

*4.2.2 Statistical Significance.* The primary objective of our analysis is to identify statistically significant variations in the performance of LLMs across various languages. We conducted an Analysis of Variance (ANOVA) for each metric to determine whether the mean performances across the languages were statistically different. As shown in Table A13, the one-way ANOVA tests for *all* metrics and *all* temperatures revealed statistically significant differences among the languages. This suggested that the performance for at least one language had statistically significant difference from the rest. For example, at $\tau = 0.0$, the $\mathcal{F}$-statistic for $\text{sim}_{\text{sent}}$ / BERTScore / $\text{sim}_{\text{1-gram}}$ were 153.47 / 157.28 / 190.94 with $p$-values of 2.52e-80 / 5.93e-82 / 0.29e-96.

In cases where the ANOVA results indicated significant differences among languages, we employed a *post-hoc Tukey Honestly Significant Difference (HSD) test* and an *unpaired t-test* to pinpoint

which particular language pairs exhibited significant disparities in performance on a given metric. As shown in Table 2, 3, the $p$-values for *English-Spanish* on $\tau = 0.0$ generally exceeded the significance level of 0.05, indicating comparable performances. In contrast, other language pairs suggest statistically significant performance variations. The results for unpaired t-test were similar, as shown in Table A12 in Appendix D.2. For MedAlpaca-30b (Table A10), we observed an increase in $\text{sim}_{\text{n-gram}}$ and decrease in topic-level consistency; however, the language disparity was less significant (details in Appendix D).

## 5 VERIFIABILITY

The last critical criterion in XLingEval is verifiability, which measures a model's capacity to authenticate the validity of claims. Within this framework, the LLM acts as a discriminator and distinguishes between correct and erroneous/irrelevant responses to a given query, in contrast to previous settings where the LLMs act as

LiveQA

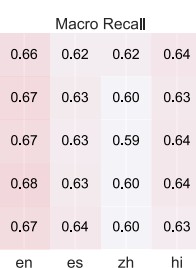 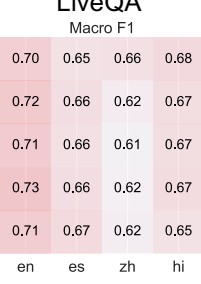 

**Figure 4: Results of LiveQA on metrics of the verifiability experiment, including macro precision, macro recall, macro F1-score, accuracy, and area under the curve (AUC). Each column represents a distinct metric. The x- and y-axis of each heatmap represents varying languages and temperatures $\tau$, respectively. The results for the other datasets are in the Appendix (Figure A3)**

generators. For example, users may rely on LLMs to corroborate the validity of their health-related knowledge. However, LLMs may produce ambiguous or contradictory responses [46, 47]. Therefore, the capability of verifiability in LLMs is crucial for streamlining mitigation strategies like Self-Debug [48] and rectifying harmful or misleading outputs [49].

XLINGEVAL 's verifiability evaluation protocol is designed as follows. The model takes as input a set of question-answer pairs $(q_E, a_E)$ for English, and $(q_{NE}, a_{NE})$ for non-English languages. It predicts a binary label $L_E$ or $L_{NE}$ about whether the response is a correct answer to the question. The question-answer pairs cover a diverse set of assertions, spanning both accurate and inaccurate claims. We then compare the model's answers to the ground truth to determine its proficiency in claim verification.

We employed slightly different settings for different datasets. In the HealthQA dataset, each question is associated with one correct answer (termed "positive example") and nine incorrect/irrelevant answers (termed "negative examples"). LiveQA and MedicationQA do not provide negative question-answer pairs. Therefore, for each question in these datasets, we randomly sampled four responses from the entire set of answers to serve as negative examples. Our evaluation employed five metrics: macro-precision, macro-recall, macro F1-score, accuracy, and Area Under the Curve (AUC). Details of the evaluation metrics are in Appendix D.1.

### 5.1 Results

Figure 4 shows the verifiability results on LiveQA across 5 temperatures $\tau$. GPT-3.5 achieved only 0.66/0.62/0.67 on the 3 non-*English* datasets, a sharp decrease compared to its performance of 0.73 in *English*. The language discrepancy is even larger on HealthQA (Figure A3), where GPT-3.5 provided comparable performances in *English* and *Spanish* but significantly worse results on *Chinese* and *Hindi*. At $\tau = 1.0$, the macro F-1 for English and Spanish were both 0.85 on HealthQA, whereas those for Chinese and Hindi were 0.73 and 0.65, respectively, reinforcing our hypothesis that LLMs' verifiability varies across languages. The AUC showed a similar pattern, with 0.92/0.87 for *English*/*Spanish* but only 0.68/0.62 for *Chinese*/*Hindi*. Meanwhile, model performance remained relatively stable across different $\tau$, suggesting that modulating the model's generative randomness does not substantially influence its ability to validate answers. As shown in Figure A11, the standard deviation of performances are lower than 0.01. In most settings, *English* and *Hindi* demonstrated the most and the least variations, respectively.

## 6 RELATED WORKS

### 6.1 Large Language Models (LLMs)

The development of language models has witnessed significant transitions from smaller-scale transformer-based models such as BERT [50], RoBERTa [51], and XLNet [52] to recent highly parameterized models, including GPT-3.5/4 [7], Bard [8], ChatGLM [53, 54], LLaMA [15], etc. These LLMs exhibit distinct capabilities in reasoning, understanding, and summarization [55–60], offering potentials in healthcare for user-friendly medical summaries and query resolutions. By generating user-friendly summaries and addressing medical inquiries, these models can significantly enhance accessibility to health-related information.

Although existing studies have demonstrated the proficiency of LLMs on medical benchmarks [25–27, 61], they do not necessarily reflect real-world human-LLM interactions. In practical scenarios, individuals often consult LLMs for symptom evaluation, health precautions, or clarifications on medical terminology. Our research seeks to address this disparity, providing insights into how well the general public can engage with and utilize these LLMs.

### 6.2 Language Disparity

Despite the proliferation of LLMs, a notable limitation challenge in the development of LLMs is the pronounced focus on English-centric models and training data [14, 62–64]. For instance, LLaMA 2 sources nearly 90% of its pretraining data from English texts [15], and a substantial portion of GPT-4's pretraining data is similarly English-centric [7]. This uneven data distribution casts doubts over the genuine multilingual capabilities of these models. Recognizing and addressing such *language disparity* in LLMs is paramount. Endeavors are being made to investigate these disparities and work towards more inclusive language models. Efforts are underway to promote more inclusive LLMs that not only improve information accessibility but also foster global health literacy. Ensuring diverse language representation is crucial not just for broadening community inclusion, but also for facilitating *diversity* and *inclusiveness* in the development and usage of LLMs, and promote equitable access to services powered by these technologies [65–67].

## 7 DISCUSSION

We presented a multi-dimensional evaluation of the cross-lingual capabilities of LLMs in the healthcare domain. Our results indicate that a consistent disparity exists between the capabilities of LLMs

in answering healthcare queries in the English language and non-English languages. We now discuss the implications of our findings.

**Equity and accessibility of healthcare information**. Large language models are advocated as language technologies that provide *accessible* healthcare information [27, 68, 69]. However, our study demonstrates that key measures relating to LLM capabilities like correctness, consistency, and verifiability are repeatedly lower for non-English languages than for the English language. As a considerable fraction of the global population is not equipped to have healthcare conversations in the English language [70], our work provides empirical evidence to raise questions about whether such claims about accessibility ignore aspects related to *equity* in language technologies in healthcare. Do the claims about accessibility of healthcare information using LLMs only apply to people who prefer to communicate in the English language?

Besides developing LLMs that provide equitable services across languages in critical domains like healthcare, which is still an open challenge, some immediate steps involve clearer communication of capabilities and potential harms. For instance, the limited capabilities of LLMs to answer healthcare-related queries, specifically in non-English languages, could be made more prominent using *trustworthiness cues*. Liao et al. [71] highlight that trustworthiness cues could empower users to make well-calibrated judgments while adopting AI technologies. Similarly, the accessibility claims relating to large language models in healthcare should be communicated while precisely mentioning the languages such capabilities were evaluated on [72]. This is particularly important as LLMs are being integrated within Web-based search frameworks (e.g., Bing Chat and Google's Generative AI Search) as a notable fraction of search queries on platforms like Google and Bing are health-related [73].

**Generalized framework for evaluating LLM Applications**. In this work, we presented a framework for assessing the efficacy of LLMs in the healthcare domain. The facet of generalizability inherent in our framework is exemplified through the evaluation carried out on two distinct LLMs, GPT-3.5 and MedAlpaca. Additionally, the criteria introduced in this work can be modified to adapt to other critical domains such as legal, finance, and education, where correctness, verifiability, and consistency of information provided by LLMs are also of major importance [69, 74–77]. It is worth emphasizing that the evaluation metrics employed in our study possess the adaptability to be directly applied to the aforementioned domains. However, it remains imperative to exercise discretion in tailoring these metrics to meet the specific requirements of each domain. For instance, in legal contexts, where considerations of legal precedence and historical case information assume paramount importance, it is necessary to introduce modifications or novel metrics within the correctness criterion to accommodate these unique domain-specific intricacies. Furthermore, as shown in our work, we highlight the need for the adoption of cross-lingual analysis in frameworks to assess the capabilities and potential harms.

**Likely causes of language disparity**. Across our evaluation metrics, we noted a disparity in LLM performance among languages. This disparity is notably more pronounced in the case of *Hindi* and *Chinese* as compared to *Spanish*. The underlying rationale for this discrepancy can be attributed primarily to two key factors: the limited availability of data resources for Non-English languages and the presence of a highly imbalanced data distribution employed in the training of the LLMs [7, 15]. The performance disparity across language is further heightened in instances involving domain-specific LLMs, access to multilingual data is difficult, as exemplified in the results pertaining to MedAlpaca in Appendix D.3. High-precision machine translation has been employed as a possible solution in past works [78, 79]. However, critical domains such as healthcare require extensive human evaluation of translation to prevent serious ramifications. A potential solution for this problem requires close collaboration with medical experts and endorsement of specific training data resources by medical/healthcare organizations.

**Future of LLMs in Healthcare**. One of the implications arising from our study centers on the discourse surrounding the future of LLMs within high-stakes domains, particularly healthcare. While a prevailing strategy focuses on the development of general-purpose LLMs with larger number of parameters trained on larger datasets [80], it is essential to acknowledge the inherent limitations of such models, including their deficiency in domain-specific knowledge and vulnerability to hallucinations [7, 81]. In contrast, domain-specific LLMs have shown promising potential and efficacy within the healthcare domain [27, 82]. However, it is critical to underscore that additional precautions and safeguards are required to mitigate the risk of adverse consequences stemming from the information generated by these models. Augmenting conversational models with knowledge bases [83], and implementing semi-automated procedures for verifying the quality of training datasets [80], emerge as prospective solutions to enhance the reliability and safety of the outputs in high-stakes domains like healthcare.

## 8 CONCLUSION AND LIMITATIONS

We presented XLINGEVAL, a holistic cross-lingual evaluation framework focusing on three fundamental criteria for LLMs — accuracy, consistency, and verifiability. We conducted an exhaustive series of automated and human evaluation experiments with four of the world's most widely spoken languages – *English*, *Chinese*, *Hindi*, and *Spanish*. The outcomes of these experiments revealed disparities inherent in LLM responses across these languages, underscoring the pressing necessity for advancements in cross-lingual capabilities. Moreover, we introduced XLINGHEALTH, an innovative cross-lingual healthcare benchmark that serves as a pivotal tool for assessing the multilingual capabilities of LLMs.

While our study represents a novel contribution to the field, it is essential to acknowledge certain limitations. Primarily, due to the unavailability of open access to a general-purpose multilingual LLM of a scale comparable to GPT-3.5, we were constrained to use a smaller healthcare-focused LLM, MedAlpaca, for comparative analysis. Additionally, our analysis was constrained by the absence of readily available multilingual datasets specific to the healthcare domain. This constraint necessitated the creation of multilingual versions through machine translation, introducing potential limitations in terms of translation quality. Overall, our research underscores the urgent imperative of enhancing the cross-lingual capabilities of these models and promoting equitable access to information across linguistic boundaries.

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

**Table A1: Statistics of the datasets. '#Words (Q)' and '#Words (A)' represent the average number of words in the questions and ground-truth answers of the datasets, respectively.**

| Dataset | #Examples | #Words (Q) | #Words (A) |
|---|---|---|---|
| HealthQA | 1,134 | 7.72 ± 2.41 | 242.85 ± 221.88 |
| LiveQA | 246 | 41.76 ± 37.38 | 115.25 ± 112.75 |
| MedicationQA | 690 | 6.86 ± 2.83 | 61.50 ± 69.44 |

**Table A2: Translation quality of the three machine translation tools utilized in this paper. We evaluate 150 examples per language. Each example is assigned 3 annotators. "C-GPT" refers to ChatGPT and "M-MT" refers to MarianMT. "Google" represents Google Translate. Texts in bold represent the best performance for the given language.**

| | Fluency | | | Meaning | | | |
|---|---|---|---|---|---|---|---|
| | C-GPT | Google | M-MT | C-GPT | Google | M-MT | Cohen's $\kappa$ |
| es | **4.38** | 4.25 | 3.89 | **4.40** | 4.12 | 3.98 | 0.86 |
| zh | **4.42** | 4.26 | 3.83 | **4.33** | 4.10 | 3.80 | 0.84 |
| hi | 4.21 | **4.36** | 3.36 | 4.32 | **4.35** | 2.87 | 0.81 |

## A DETAILS OF DATASET CONSTRUCTION

Observing the lack of existing multilingual QA datasets in healthcare domains, we curate a novel benchmark. To ensure the quality of the dataset, we conduct a human evaluation on the translation quality of three popular approaches commonly adopted in translating academic documents: Google Translate [84], MarianMT [85], and ChatGPT [6]. To comprehensively evaluate the capability of each model in translating different datasets, we randomly selected 50 questions from each dataset, resulting in a total of 150 questions. Our evaluation of translation quality aligns with established standards in previous works [14]. A total of 450 translation pairs (150 questions across 3 languages) were evaluated. Each example was reviewed by three independent annotators who scored the translations using a five-point Likert scale (1: strongly disagree — 5: strongly agree) on two critical dimensions:

(1) **Fluency**. Is the [TARGET LANGUAGE] version a good translation of the English text?
(2) **Meaning**. Does the [TARGET LANGUAGE] version faithfully convey the same meaning as the English text?

From Table A2, it can be noted that our evaluation revealed ChatGPT to outperform other approaches in translations from English to both Chinese and Spanish, while Google Translate exhibits superior performance in English-to-Hindi translation. Thus, for optimal results in each non-English language, we harnessed the best-performing model to achieve the highest translation quality.

## B RATIONALE FOR THE CORRECTNESS CRITERIA

We merged and modified the categories from the past work [27] to create two consolidated axes for the comparative evaluations of the answers produced by LLMs with the Ground Truth answer. The two proposed axes cover three essential dimensions – contradiction, appropriateness, and comprehensiveness. The dimension of contradiction addresses situations wherein LLMs' responses exhibit inconsistencies compared to the answers provided by medical experts, signifying inaccuracies in the LLM-generated responses. While the LLM answer may not specifically contradict the Ground Truth answer, it may still be irrelevant to the asked question. We check this scenario through asking to evaluate the similarity between the LLM and the Ground Truth answer, keeping the contextual relevance to the original question through *Phase 2* prompting. If the LLM-generated answer is determined to be similar to the Ground Truth answer while keeping contextual alignment with the question, it is considered appropriate. Finally, if both answers are evaluated as similar and appropriate, then we compare the comprehensiveness of both answers through the last step in the *Phase 2* prompt.

## C HUMAN EVALUATION

### C.1 Annotation Platform

Figure A1 presents the different pages from the annotation platform designed for conducting the human evaluation for the Correctness experiment. Each instance within the annotation dataset comprised a quadruple, consisting of a question, an expert-curated answer, a response generated by the GPT-3.5 model, and a reasoning generated from *Phase-2* prmpting that elucidated the justification behind the classification label ascribed to the given example. The annotators needed to answer a yes/no based question on whether they agreed with the reasoning and classification label provided by GPT-3.5. Additionally, in cases where the annotator did not agree with the reasoning, we asked them to provide the reasoning for selecting the 'no' option and along with reporting the correct relationship between the two answers. We assigned the majority label to each instance based on the annotations from three annotators.

### C.2 Results

Table A7 presents the correlation number for each batch for *English (en)*, *Spanish (es)*, *Chinese (zh)*, and *Hindi (hi)*. Each batch for *English*, and *Spanish* was annotated by three medical experts. On the other hand, each batch for *Chinese* was annotated by two annotators and each batch for *Hindi* was annotated by 1 annotator. It is worth noting that a comprehensive, multi-faceted approach was adopted in the recruitment of participants for this annotation task, encompassing a wide range of sources such as crowdsourcing platforms, social media platforms, and offline channels. Despite these efforts, hiring additional medical experts proficient in *Chinese* or *Hindi* proved challenging. We observed a high correlation between the automated and human labels in the annotation dataset in each language with more than 90% agreement for each of *English*, *Spanish*, and *Chinese*.

## D AUTOMATED EVALUATION

### D.1 Evaluation Metrics in Verifiability

As described in Section 5, we use five metrics in the verifiability experiments, including macro precision, macro recall, macro F1-score, Accuracy, and the Area Under the Curve (AUC).

**Table A3: Prompts used in the experiments. Question refers to the question from the dataset, Answer 1 and Answer 2 refers to Ground truth and LLM answer respectively.**

---

*Correctness Prompt (Phase 1)*

You are an expert in medicine, health, and pharmaceutics. Please answer this health- and medical-related queries in [LANGUAGE]. Your answers should be in one or more paragraphs without listing points/lists.
**Question**: [QUESTION]

---

*Correctness Prompt (Phase 2)*

Given below is the question and two answers written in [LANGUAGE] for the question.
**Question**: [QUESTION]
**Answer 1**: [ANSWER 1]
**Answer 2**: [ANSWER 2]

Compare Answer 2 with Answer 1 on the basis of answer similarity, comprehensiveness and appropriateness.
Answer 2 contradicts Answer 1 if Answer 2 provides any information that contradicts the information provided in Answer 1 and Answer 2 incorrectly answers the Question.
Answer 2 is neither contradictory nor similar if Answer 2 answers the Question but Answer 1 and Answer 2 are dissimilar. First, evaluate whether Answer 2 provides similar information, contradictory information as compared to Answer 1. Support the evaluation with a reasoning. Only if both answers are similar, evaluate the comprehensiveness and appropriateness of Answer 2 in comparison to Answer 1 and provide a reasoning for it.
Finally, based on the previous evaluation, choose one option from the following option:

1) Answer 2 provides neither contradictory nor similar information in comparison to Answer 1
2) Answer 2 provides contradictory information compared to Answer 1
3) Answer 2 provides more comprehensive and appropriate information.
4) Answer 2 provides less comprehensive and appropriate information

Your answer should first provide all reasoning followed by the chosen option string in the last line. The chosen option string should be an exact match with one of the given options.

---

*Consistency Prompt 1*

You are an expert in medicine, health, and pharmaceutics. Please answer the following medical question or give your medical diagnosis based on the following scenario.
**Answer**: [ANSWER]

---

*Verifiability Prompt 1*

You are an expert in medicine, health, and pharmaceuticals. Respond to me whether the following response is a correct answer (or diagnosis) to the following question (or scenario).
**Question**: [QUESTION]
**Answer**: [ANSWER]

---

**Macro precision** and **macro recall** are the average precision and recall across all classes, computed as:

$$P_{macro} = \frac{1}{n} \sum_{i=1}^{n} \frac{TP_i}{TP_i + FP_i}, \quad (4)$$

$$R_{macro} = \frac{1}{n} \sum_{i=1}^{n} \frac{TP_i}{TP_i + FN_i}, \quad (5)$$

where $n$ is the number of classes, $TP_i$, $FP_i$, $FN_i$ are the number of true positives, false positives, and false negatives for class $i$, respectively.

**Macro F1-score** is the harmonic mean of macro precision and macro recall, computed as:

$$F1_{macro} = 2 \frac{P_{macro} \cdot R_{macro}}{P_{macro} + R_{macro}}. \quad (6)$$

**Accuracy** is the percentage of correctly predicted examples among all examples:

$$Acc = \frac{TP + TN}{TP + TN + FP + FN}, \quad (7)$$

where TP, FP, TN, FN are the number of true positives, false positives, true positives, and false negatives, respectively.

**AUC**, or Area Under the ROC Curve, signifies the performance of the classification model across all thresholds. It measures the two-dimensional area underneath the ROC curve (receiver operating characteristic curve). An ROC curve plots the True Positive Rate (TPR) against the False Positive Rate (FPR):

$$TPR = \frac{TP}{TP + FN}, \quad (8)$$

$$FPR = \frac{FP}{FP + TN}. \quad (9)$$

## D.2 Results of Unpaired t-tests

Due to varying content filtering criteria on each language, GPT-3.5 usually refuses to answer a different set of questions on each language. We thus supplemented our analysis with an unpaired t-test (Table A12). Using a significance threshold ($\alpha$) of 0.05, the $p$-values show that for most metrics, the English-Spanish comparison at $\tau = 0.0$ is statistically consistent ($p > \alpha$). However, significant cross-lingual differences emerge when $\tau$ increases towards 1.0. For all other language pairings, the $p$-values are consistently lower than $\alpha$, revealing statistically significant performance discrepancies and language disparities. Analogous patterns were noted across other datasets and temperatures. Similar results are observed on the rest of the datasets and temperatures.

## D.3 Results on MedAlpaca

When deploying LLM-based conversational agents, a primary consideration arises: Is it more effective to deploy a larger, general LLM or a smaller, specialized model to respond to user queries? This section delves into this question by examining MedAlpaca [18], a specialized LLM tailored for the medical domain. MedAlpaca is fine-tuned from LLaMA [15] using the Medical Meadow dataset [18], and has demonstrated exceptional performances on the United States Medical Licensing Examination (USMLE). For our assessment, we focus on its largest version: MedAlpaca-30b.

*D.3.1 Consistency.* From Table A10, we note a significant decline in topical-level consistency for MedAlpaca compared with GPT-3.5 (Table A9). Meanwhile, its lexical consistency is superior, as demonstrated by its higher $\text{sim}_{\text{1-gram}}$, $\text{sim}_{\text{2-gram}}$.

Given the propensity of smaller-scale LLMs to generate responses predominantly in English, we introduced a new metric, we introduce a new metric, **language consistency**, to gauge the alignment of the response language to the source sentence's language. For an example target language $l$ and a this metric determines the fraction of sentences generated in the target languages by the LLM relative to all sentences:

$$\text{LangCons}(q, l) = \sum_{s_i \in S_q} \frac{\text{Count}(s_i, l))}{\sum_{l'} \text{Count}(s_i, l')} \qquad (10)$$

where $l, l'$ are languages, $s_i$ is a generated answer, and $S$ is the set of all generated answers to the question $q$ Empirically, we use the langid[2] package to determine the language of each sentence. We calculate the aggregated *language consistency* by averaging over all examples in Table A4.

We found that MedAlpaca-30b has the lowest language consistency in Spanish. Also, language consistency varies across different temperatures.

*D.3.2 Verifiability.* As shown in Table A14, MedAlpaca-30b does not demonstrate good performances in authenticating claims. It demonstrates a highly imbalanced prediction towards the negative class, leading to low macro-precision/recall/F1 and AUC scores in most languages.

*D.3.3 Correctness.* The correctness results for MedAlpaca-30b is shown in Table A8. As observed, there is a sharp decrease in the number of answers where MedAlpaca produces a more comprehensive and appropriate answer as compared to Ground Truth. For HealthQA, we observed a relative decrease of $\sim$92.23%, $\sim$97.93%, and $\sim$93.26% for *Spanish*, *Chinese*, and *Hindi* respectively as compared to *English*. parallel trend was observed for the LiveQA and Medication QA datasets. In contrast to GPT-3.5 results, we observed the majority proportion of answers in non-English languages being assigned the 'Neither contradictory nor similar' label. This observation stems from the fact that MedAlpaca either did not produce the answer in the respective language or produced a hallucinated answer with repeated tokens.

## D.4 Content Filtering

GPT-3.5/4 leverages an additional post-processing step of content filtering to ensure content safety and relevance. This is important in the medical domain as end users have varied levels of medical knowledge and are potentially subject to misunderstandings or misapplications. Table shows the content filtering percentage in the verifiability experiments. The results indicate that all languages except Spanish exhibit little variation in content filtering iwith respect to temperature $\tau$. For Spanish, GPT-3.5 filtered 0.2%/0.5% on temperatures 0.0/1.0, respectively. It is worth noting that the model consistently recorded a zero filtering rate for Chinese (zh), suggesting possible vulnerabilities to generating inappropriate content in Chinese contexts.

**Table A4: Language consistency of the medAlpaca-30B model. The language consistency generally decreases as the temperature $\tau$ increases, except for Spanish.**

| $\tau$ | en | es | zh | hi |
|--------|-------|-------|-------|-------|
| 0.0 | 99.85 | 24.02 | 79.62 | 81.52 |
| 1.0 | 99.45 | 32.62 | 70.54 | 55.35 |

**Table A5: Percentage of examples filtered by GPT-3.5/4. Content filtering rates remain consistent across temperatures except for Spanish. Additionally, Chinese (zh) demonstrates minimal content filtering.**

| Model | $\tau$ | en | es | zh | hi |
|---------|--------|------|------|------|------|
| GPT-3.5 | 0.0 | 0.2% | 0.2% | 0.0% | 0.3% |
|         | 1.0 | 0.2% | 0.5% | 0.0% | 0.3% |
| GPT-4   | 0.0 | 0.2% | 0.2% | 0.0% | 0.3% |
|         | 1.0 | 0.2% | 0.2% | 0.0% | 0.3% |

**Table A6: Notations used in this paper.**

| Notation | Description |
|----------|-------------|
| $w_i, \mathbf{w}_i$ | a token and its contextualized embedding |
| $q_{\text{E}}, q_{\text{NE}}$ | A question in English / non-English languages |
| $a_{\text{E}}, a_{\text{NE}}$ | An answer in English / non-English languages |
| $s_i$ | a response generated by LLM |
| $D, |D|$ | A dataset and the number of examples it contains. |
| $|s_i|$ | Number of words in the response $s_i$ |
| $l$ | a language |

---

[2]https://github.com/saffsd/langid.py

revised 12 March 2009; accepted 5 June 2009

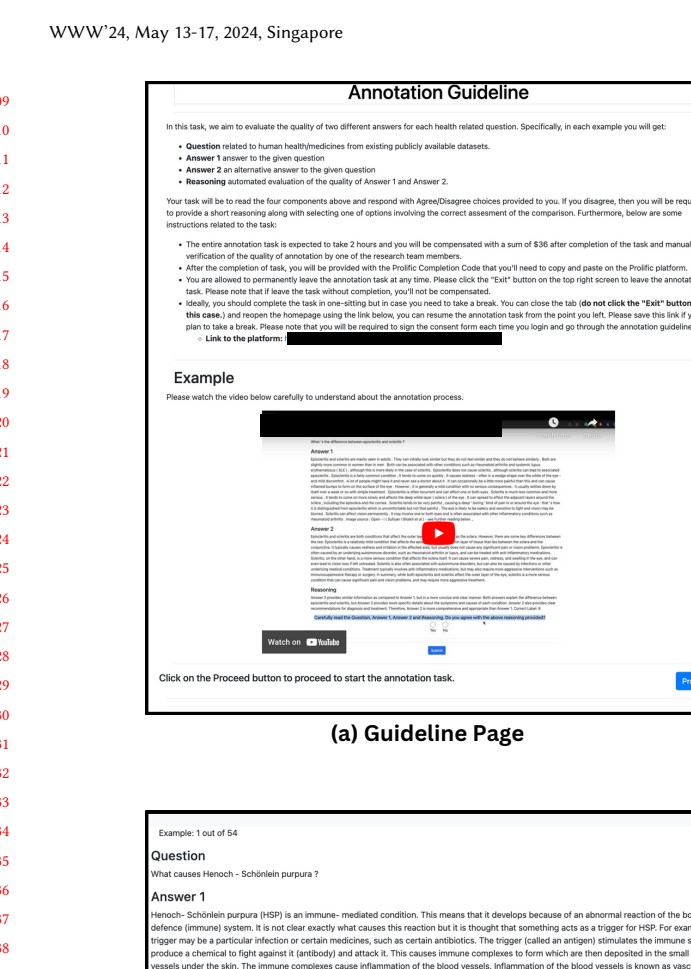

**(a) Guideline Page**

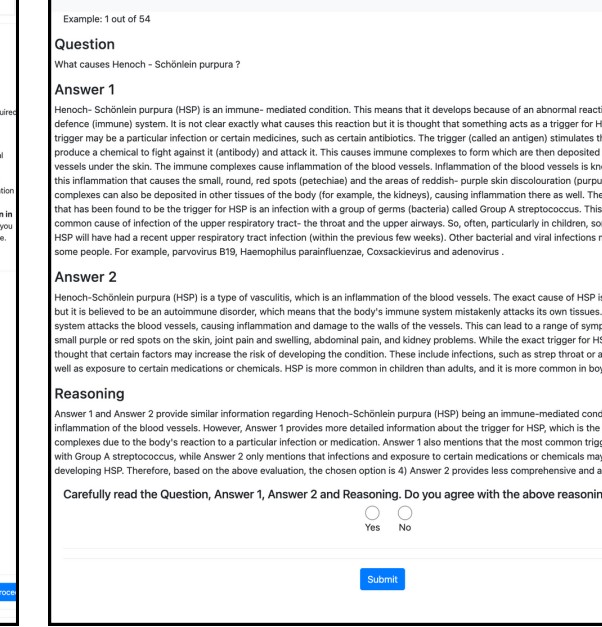

**(b) Annotation Example with Question, LLM Answer, Ground Truth Answer, and Reasoning generated in Phase 2 prompting**

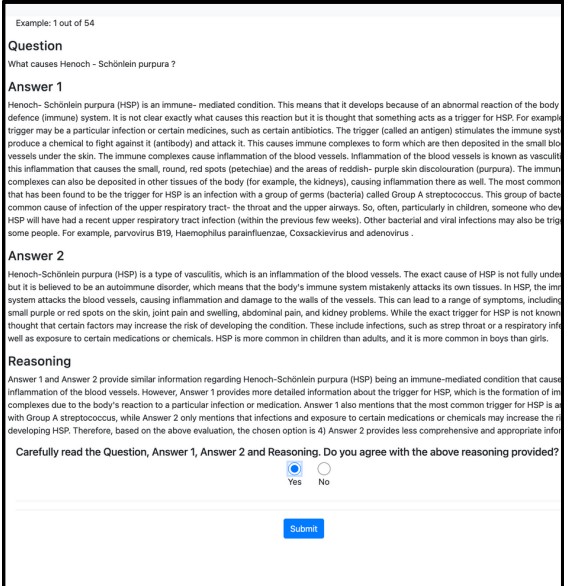

**(c) Case when the annotator agrees with the reasoning**

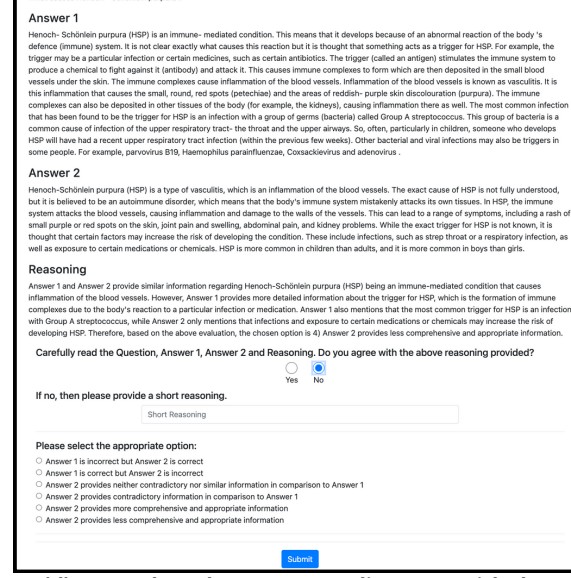

**(d) Case when the annotator disagrees with the reasoning**

Figure A1: Annotation Platform created for the human evaluation for Correctness experiment.

**Table A7: Human Evaluation Results for Correctness metric. *** denotes annotations performed with three annotators, ** denotes annotations denotes annotations performed with two annotators, and * denotes annotations denotes annotations performed with one annotator.**

| Metric Type | en (batch-1) | en (batch-2) | es (batch-1) | es (batch-2) | zh (batch-1) | zh (batch-2) | hi (batch-1) | hi (batch-2) |
|---|---|---|---|---|---|---|---|---|
| Correlation (Automated & Majority Human Label) | 96.08%*** | 92.31%*** | 94.12%*** | 96.15%*** | 70.59%** | 84.62%** | 84.31%* | 84.62%* |

**Table A8: Automated Correctness evaluation across four languages: English (en), Spanish (es), Chinese (zh), and Hindi (hi) for MedAlpaca-30b.**

| Information Comparison (LLM Answer vs Ground Truth Answer) | HealthQA | | | | LiveQA | | | | MedicationQA | | | |
|---|---|---|---|---|---|---|---|---|---|---|---|---|
| | en | es | zh | hi | en | es | zh | hi | en | es | zh | hi |
| More comprehensive and appropriate | 193 | 15 | 4 | 13 | 58 | 7 | 5 | 15 | 131 | 20 | 15 | 12 |
| Less comprehensive and appropriate | 498 | 199 | 112 | 106 | 60 | 13 | 41 | 55 | 121 | 55 | 61 | 68 |
| Neither contradictory nor similar | 318 | 737 | 738 | 843 | 93 | 194 | 168 | 132 | 333 | 489 | 482 | 502 |
| Contradictory | 124 | 182 | 277 | 172 | 34 | 32 | 32 | 44 | 105 | 126 | 131 | 108 |
| No Response | 1 | 1 | 3 | - | 1 | - | - | - | - | - | 1 | - |

**Table A9: Performance comparison of consistency experiments on GPT-3.5 across varying languages. We show the average performances over different temperatures ($\tau$) and their performance drop (in percentage) compared to English.**

| Med | $\text{sim}_{\text{sent}}$ | BERTScore | $\text{sim}_{\text{1-gram}}$ | $\text{sim}_{\text{2-gram}}$ | Length | $\text{sim}_{\text{HDP}}$ | $\text{sim}_{\text{LDA}}^{20}$ | $\text{sim}_{\text{LDA}}^{100}$ |
|---|---|---|---|---|---|---|---|---|
| en | 0.9699/0.0% | 0.7040/0.0% | 0.5201/0.0% | 0.3533/0.0% | 109.0798/0.0% | 0.9256/0.0% | 0.9183/0.0% | 0.8694/0.0% |
| es | 0.9677/-0.2% | 0.6905/-1.9% | 0.5016/-3.5% | 0.3328/-5.8% | 100.9373/-7.5% | 0.9204/-0.6% | 0.9094/-1.0% | 0.8562/-1.5% |
| zh | 0.9602/-1.0% | 0.6408/-9.0% | 0.4315/-17.0% | 0.2647/-25.1% | 106.6152/-2.3% | 0.8910/-3.7% | 0.8747/-4.7% | 0.8013/-7.8% |
| hi | 0.9395/-3.1% | 0.5797/-17.7% | 0.3717/-28.5% | 0.2009/-43.1% | 78.3874/-28.1% | 0.8589/-7.2% | 0.7762/-15.5% | 0.6490/-25.4% |

| Heal | $\text{sim}_{\text{sent}}$ | BERTScore | $\text{sim}_{\text{1-gram}}$ | $\text{sim}_{\text{2-gram}}$ | Length | $\text{sim}_{\text{HDP}}$ | $\text{sim}_{\text{LDA}}^{20}$ | $\text{sim}_{\text{LDA}}^{100}$ |
|---|---|---|---|---|---|---|---|---|
| en | 0.9755/0.0% | 0.7013/0.0% | 0.5188/0.0% | 0.3476/0.0% | 131.3095/0.0% | 0.9104/0.0% | 0.9485/0.0% | 0.9342/0.0% |
| es | 0.9722/-0.3% | 0.6858/-2.2% | 0.4976/-4.1% | 0.3253/-6.4% | 119.3215/-9.1% | 0.9089/-0.2% | 0.9421/-0.7% | 0.9269/-0.8% |
| zh | 0.9671/-0.9% | 0.6368/-9.2% | 0.4187/-19.3% | 0.2493/-28.3% | 134.9392/2.8% | 0.8817/-3.2% | 0.9233/-2.7% | 0.9032/-3.3% |
| hi | 0.9428/-3.4% | 0.5537/-21.0% | 0.3412/-34.2% | 0.1715/-50.7% | 96.6498/-26.4% | 0.8055/-11.5% | 0.8378/-11.7% | 0.7940/-15.0% |

| Live | $\text{sim}_{\text{sent}}$ | BERTScore | $\text{sim}_{\text{1-gram}}$ | $\text{sim}_{\text{2-gram}}$ | Length | $\text{sim}_{\text{HDP}}$ | $\text{sim}_{\text{LDA}}^{20}$ | $\text{sim}_{\text{LDA}}^{100}$ |
|---|---|---|---|---|---|---|---|---|
| en | 0.9706/0.0% | 0.6631/0.0% | 0.4798/0.0% | 0.3060/0.0% | 146.8889/0.0% | 0.8913/0.0% | 0.9237/0.0% | 0.8784/0.0% |
| es | 0.9674/-0.3% | 0.6461/-2.6% | 0.4600/-4.1% | 0.2831/-7.5% | 136.8197/-6.9% | 0.9111/2.2% | 0.9229/-0.1% | 0.8354/-4.9% |
| zh | 0.9613/-1.0% | 0.6015/-9.3% | 0.3996/-16.7% | 0.2229/-27.2% | 144.7613/-1.4% | 0.8565/-3.9% | 0.8774/-5.0% | 0.8000/-8.9% |
| hi | 0.9415/-3.0% | 0.5339/-19.5% | 0.3329/-30.6% | 0.1515/-50.5% | 104.9724/-28.5% | 0.8170/-8.3% | 0.7672/-16.9% | 0.5979/-31.9% |

**Table A10: Performance comparison of consistency experiments on MedAlpaca-30b across varying languages and the performance drop compared to English.**

| Live | $\text{sim}_{\text{sent}}$ | BERTScore | $\text{sim}_{\text{1-gram}}$ | $\text{sim}_{\text{2-gram}}$ | Length | $\text{sim}_{\text{HDP}}$ | $\text{sim}_{\text{LDA}}^{20}$ | $\text{sim}_{\text{LDA}}^{100}$ |
|---|---|---|---|---|---|---|---|---|
| en | 0.8738/0.0% | 0.7649/0.0% | 0.5427/0.0% | 0.4967/0.0% | 84.1697/0.0% | 0.8210/0.0% | 0.6636/0.0% | 0.5521/0.0% |
| es | 0.8517/-2.5% | 0.7549/-1.3% | 0.5585/2.9% | 0.5136/3.4% | 91.3254/8.5% | 0.7659/-6.7% | 0.6565/-1.1% | 0.5808/5.2% |
| zh | 0.8584/-1.8% | 0.7507/-1.9% | 0.5373/-1.0% | 0.4955/-0.2% | 94.0495/11.7% | 0.8619/5.0% | 0.6847/3.2% | 0.5528/0.1% |
| hi | 0.8469/-3.1% | 0.7424/-2.9% | 0.5368/-1.1% | 0.4924/-0.9% | 68.7502/-18.3% | 0.7611/-7.3% | 0.5989/-9.7% | 0.5361/-2.9% |

**Table A11: Average *verifiability* performances on GPT-3.5 across five temperatures and their standard deviation. English (en) and Spanish (es) performances are consistently better than Chinese (zh) and Hindi (hi). The performance variations across languages are minimal, with Hindi showing the most significant variations.**

| | | $P_{macro}$ | $R_{macro}$ | $F1_{macro}$ | Accuracy | AUC |
|---|---|---|---|---|---|---|
| en | | 0.9447 + 0.0012 | 0.8113 + 0.0039 | 0.8581 + 0.0033 | 0.9220 + 0.0015 | 0.8113 + 0.0039 |
| es | | 0.9422 + 0.0012 | 0.8769 + 0.0018 | 0.9048 + 0.0015 | 0.9434 + 0.0008 | 0.8769 + 0.0018 |
| zh | | 0.8590 + 0.0028 | 0.6739 + 0.0026 | 0.7143 + 0.0031 | 0.8604 + 0.0011 | 0.6739 + 0.0026 |
| hi | | 0.8606 + 0.0079 | 0.6874 + 0.0039 | 0.7289 + 0.0049 | 0.8645 + 0.0023 | 0.6874 + 0.0039 |
| en | | 0.8119 + 0.0028 | 0.9222 + 0.0042 | 0.8552 + 0.0012 | 0.9383 + 0.0010 | 0.9222 + 0.0042 |
| es | | 0.8297 + 0.0040 | 0.8623 + 0.0067 | 0.8449 + 0.0052 | 0.9414 + 0.0017 | 0.8623 + 0.0067 |
| zh | | 0.8396 + 0.0017 | 0.6802 + 0.0013 | 0.7289 + 0.0010 | 0.9246 + 0.0002 | 0.6802 + 0.0013 |
| hi | | 0.7092 + 0.0224 | 0.6314 + 0.0334 | 0.6541 + 0.0145 | 0.9119 + 0.0192 | 0.6314 + 0.0334 |
| en | | 0.9111 + 0.0020 | 0.6701 + 0.0072 | 0.7140 + 0.0087 | 0.8649 + 0.0028 | 0.6701 + 0.0072 |
| es | | 0.9050 + 0.0039 | 0.6290 + 0.0053 | 0.6622 + 0.0072 | 0.8504 + 0.0020 | 0.6290 + 0.0053 |
| zh | | 0.9076 + 0.0031 | 0.6035 + 0.0121 | 0.6261 + 0.0174 | 0.8410 + 0.0047 | 0.6035 + 0.0121 |
| hi | | 0.8475 + 0.0076 | 0.6354 + 0.0065 | 0.6656 + 0.0092 | 0.8373 + 0.0061 | 0.6354 + 0.0065 |

**Table A12: Unpaired t-test results on English (en), Spanish (es), Chinese (zh), and Hindi (hi) on the LiveQA dataset with $\tau = 0.0$ and 1.0. $t$ and $p$ stands for the $t$-statistic and $p$-value, respectively. Asterisks (*) denotes the significance level. '*' indicates p<0.05. '**' indicates p<0.01. '***' indicates p<0.001.**

| $\tau = 0.0$ | | $sim_{BERT}$ | | BERTScore | | $sim_{1gram}$ | | $sim_{2grams}$ | | $sim_{LDA}^{20}$ | | $sim_{HDP}$ | |
|---|---|---|---|---|---|---|---|---|---|---|---|---|---|
| Language | | $t$ | $p$ | $t$ | $p$ | $t$ | $p$ | $t$ | $p$ | $t$ | $p$ | $t$ | $p$ |
| en | es | 0.83 | 4.07e-01 | 0.79 | 4.30e-01 | 0.83 | 4.04e-01 | 1.04 | 2.97e-01 | 0.63 | 5.27e-01 | 2.66 | 8.13e-03** |
| en | zh | 7.62 | 1.47e-13*** | 8.54 | 2.07e-16*** | 9.27 | 7.98e-19*** | 9.57 | 7.43e-20*** | 4.47 | 1.00e-05*** | 5.90 | 7.13e-09*** |
| en | hi | 16.19 | 8.86e-47*** | 18.75 | 2.22e-58*** | 21.31 | 3.34e-70*** | 22.37 | 4.67e-75*** | 9.47 | 1.63e-19*** | 9.07 | 3.85e-18*** |
| es | zh | 7.10 | 4.92e-12*** | 7.81 | 3.98e-14*** | 8.53 | 2.24e-16*** | 8.71 | 5.88e-17*** | 3.75 | 1.98e-04*** | 3.52 | 4.71e-04*** |
| es | hi | 15.87 | 2.41e-45*** | 18.03 | 4.33e-55*** | 20.62 | 5.64e-67*** | 21.67 | 7.93e-72*** | 8.79 | 3.13e-17*** | 6.77 | 4.13e-11*** |
| zh | hi | 9.96 | 2.93e-21*** | 9.70 | 2.49e-20*** | 11.00 | 4.29e-25*** | 11.42 | 1.03e-26*** | 5.33 | 1.55e-07*** | 3.19 | 1.51e-03** |
| $\tau = 1.0$ | | $sim_{BERT}$ | | BERTScore | | $sim_{1gram}$ | | $sim_{2grams}$ | | $sim_{LDA}^{20}$ | | $sim_{HDP}$ | |
| Language | | $t$ | $p$ | $t$ | $p$ | $t$ | $p$ | $t$ | $p$ | $t$ | $p$ | $t$ | $p$ |
| en | es | 4.58 | 6.03e-06*** | 5.40 | 1.04e-07*** | 5.86 | 8.86e-09*** | 5.93 | 5.93e-09*** | 1.33 | 1.84e-01 | 0.03 | 9.76e-01 |
| en | zh | 7.37 | 7.40e-13*** | 9.96 | 2.37e-21*** | 14.22 | 1.78e-38*** | 12.81 | 1.75e-32*** | 7.21 | 2.28e-12*** | 3.77 | 1.85e-04*** |
| en | hi | 18.02 | 1.13e-55*** | 28.94 | 8.01e-107*** | 31.23 | 4.16e-117*** | 28.71 | 8.70e-106*** | 18.87 | 1.24e-59*** | 6.96 | 1.11e-11*** |
| es | zh | 3.85 | 1.36e-04*** | 5.42 | 9.54e-08*** | 9.53 | 8.49e-20*** | 8.27 | 1.38e-15*** | 6.20 | 1.24e-09*** | 3.92 | 1.00e-04*** |
| es | hi | 15.54 | 2.58e-44*** | 25.30 | 5.03e-90*** | 28.54 | 5.52e-105*** | 28.49 | 8.68e-105*** | 18.10 | 4.75e-56*** | 7.33 | 1.03e-12*** |
| zh | hi | 10.93 | 6.06e-25*** | 17.06 | 3.07e-51*** | 16.72 | 1.12e-49*** | 19.27 | 1.58e-61*** | 11.54 | 2.65e-27*** | 2.82 | 4.95e-03** |

**Table A13: The $\mathcal{F}$-statistics and the $p$-values of ANOVA on the LiveQA dataset. For all metrics, ANOVA shows statistically significant differences between the mean performances on each metric.**

| $\tau$ | Metric | $sim_{sent}$ | BERTScore | $sim_{1-gram}$ | $sim_{2-gram}$ | length | $sim_{LDA20}$ | $sim_{HDP}$ |
|---|---|---|---|---|---|---|---|---|
| 0.0 | $\mathcal{F}$ | 153.47 | 157.28 | 190.94 | 201.70 | 35.04 | 47.08 | 33.13 |
| | $p$ | 2.52e-80 | 5.93e-82 | 8.29e-96 | 4.85e-100 | 2.01e-21 | 2.87e-28 | 2.56e-20 |
| 0.25 | $\mathcal{F}$ | 166.37 | 160.62 | 199.13 | 195.95 | 26.95 | 82.49 | 15.83 |
| | $p$ | 7.20e-86 | 1.92e-83 | 3.91e-99 | 6.99e-98 | 1.06e-16 | 3.57e-47 | 4.87e-10 |
| 0.5 | $\mathcal{F}$ | 169.11 | 199.40 | 252.44 | 253.87 | 66.61 | 109.25 | 72.64 |
| | $p$ | 7.35e-87 | 4.79e-99 | 1.04e-118 | 3.27e-119 | 6.90e-39 | 2.77e-60 | 4.68e-42 |
| 0.75 | $\mathcal{F}$ | 36.62 | 41.68 | 64.89 | 76.60 | 12.06 | 31.86 | 7.67 |
| | $p$ | 2.76e-17 | 5.11e-19 | 7.59e-26 | 9.26e-29 | 5.00e-07 | 1.48e-15 | 9.14e-05 |
| 1.0 | $\mathcal{F}$ | 149.11 | 304.02 | 368.81 | 329.65 | 20.21 | 178.26 | 22.48 |
| | $p$ | 3.75e-79 | 3.08e-138 | 1.12e-158 | 1.44e-146 | 1.06e-12 | 1.22e-91 | 4.55e-14 |

**Table A14: Results of verifiability experiments on MedAlpaca-30b.**

|     | macro_precision | macro_recall | macro_f1 | accuracy | auc |
|-----|----------------|--------------|----------|----------|-----|
| en  | 0.4538 ± 0.0793 | 0.4998 ± 0.0097 | 0.4717 ± 0.0445 | 0.7638 ± 0.0322 | 0.4998 ± 0.0097 |
| es  | 0.4983 ± 0.0423 | 0.4999 ± 0.0192 | 0.4844 ± 0.0293 | 0.7524 ± 0.0270 | 0.4999 ± 0.0192 |
| zh  | 0.5080 ± 0.0162 | 0.5033 ± 0.0116 | 0.4677 ± 0.0535 | 0.5964 ± 0.2033 | 0.5033 ± 0.0116 |
| hi  | 0.4878 ± 0.1937 | 0.4953 ± 0.0271 | 0.4429 ± 0.0550 | 0.7381 ± 0.0851 | 0.4953 ± 0.0271 |

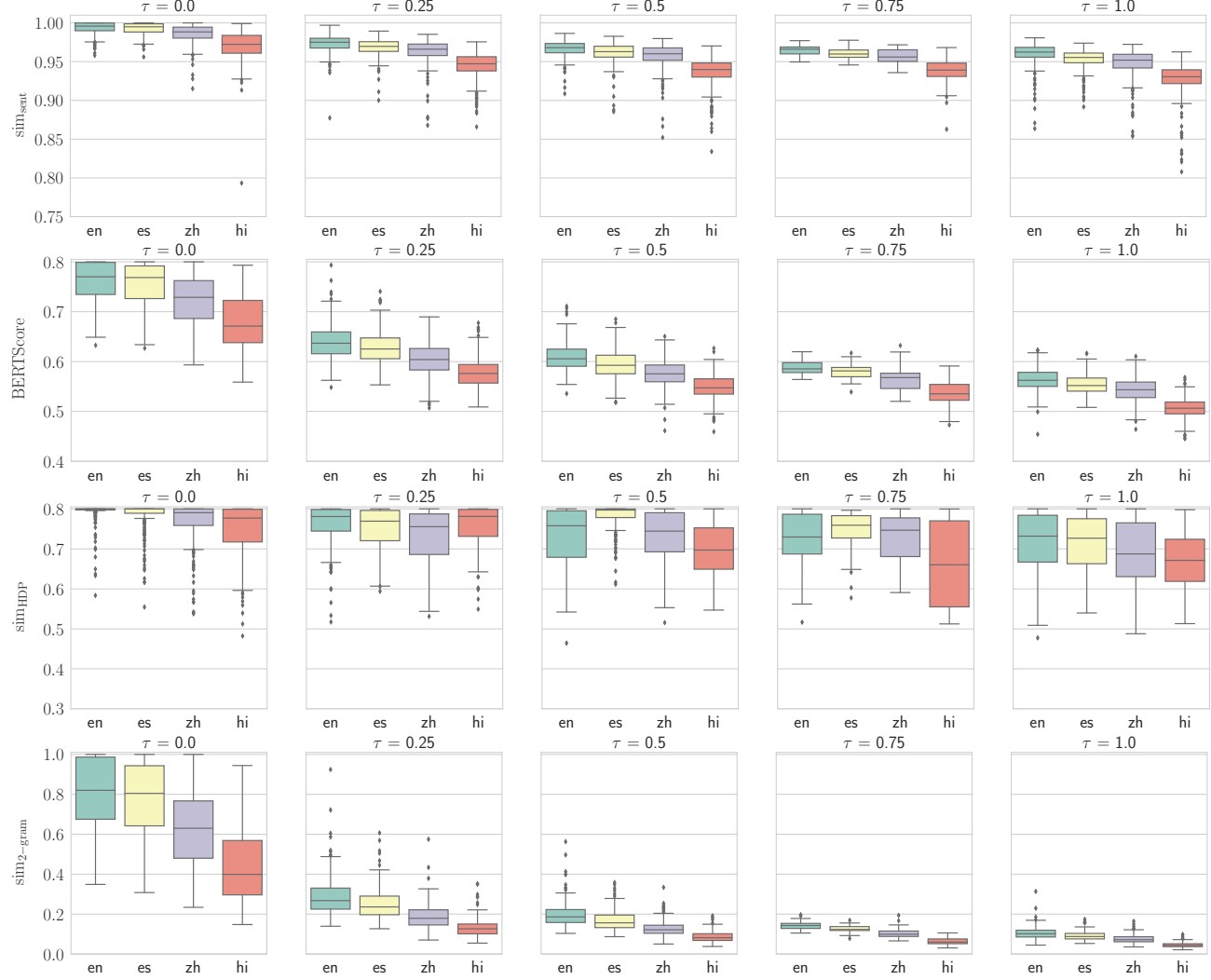

**Figure A2: Comparison of** $\text{sim}_\text{sent}$**, BERTScore,** $\text{sim}_\text{HDP}$**, and** $\text{sim}_\text{2-gram}$ **on the LiveQA dataset across 5 temperatures ($\tau$) and 4 languages.**

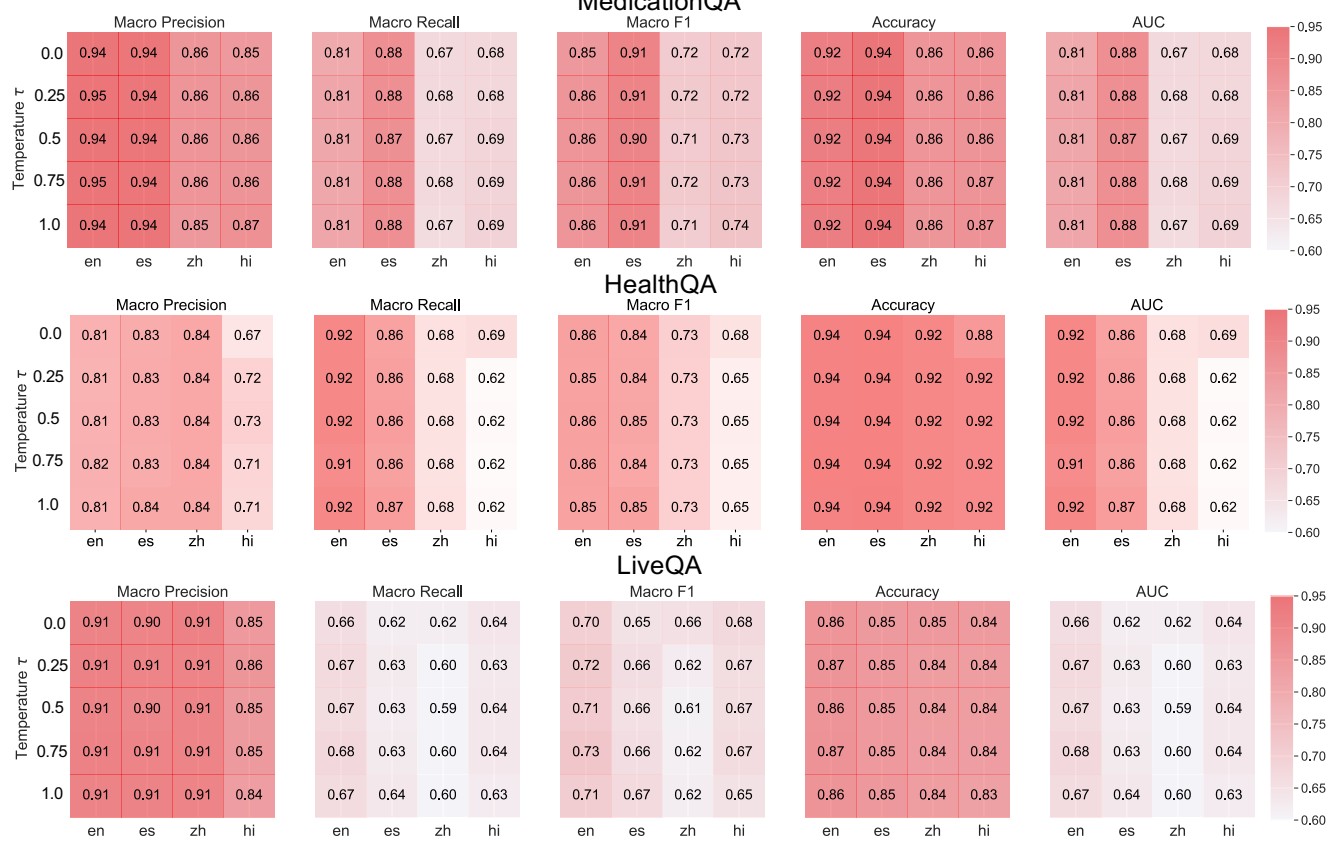

**Figure A3: Results of HealthQA, LiveQA, and MedicationQA on metrics of the verifiability experiment, including macro precision, macro recall, macro F1-score, accuracy, and area under the curve (AUC). Each column represents a distinct metric. The x- and y-axis of each heatmap represent varying languages and temperatures $\tau$, respectively.**

