# OpenReview forum: "Ask Me in English Instead: Cross-Lingual Evaluation of Large Language Models for Healthcare Queries"
_ACM.org/TheWebConf/2024/Conference — TheWebConf24 Oral_

### Official Review · Reviewer_XR35 · 2023-11-22

**Novelty:** 4
**Technical Quality:** 4

**Review:**

This paper aims to investigate the cross-lingual effectiveness of large language models (LLMs) in healthcare queries. The authors emphasize the crucial role of LLMs in transforming how the general public accesses and utilizes information, especially in critical sectors like healthcare, where people increasingly use LLMs as conversational agents for everyday queries. Despite the impressive language understanding and generation capabilities of LLMs, concerns about their safety remain paramount in these high-risk domains.

The paper notes that the development of LLMs is primarily focused on English, and their performance in non-English language contexts remains unclear, posing a critical gap in ensuring fairness in the practical application of these systems. To address this, the paper provides a framework for evaluating LLMs as multilingual dialogue systems for healthcare queries. The proposed framework, XlingEval, focuses on three fundamental criteria for evaluating LLM responses to naturalistic human-authored health-related questions: correctness, consistency, and verifiability.

Through extensive experiments in four major global languages (English, Spanish, Chinese, and Hindi), spanning three expert-annotated large health Q&A datasets, and employing a combination of algorithmic and human-evaluation strategies, the authors found a pronounced disparity in LLM responses across these languages, indicating a need for enhanced cross-lingual capabilities. The authors further propose XLingHealth, a cross-lingual benchmark for examining the multilingual capabilities of LLMs in the healthcare context. The findings underscore the pressing need to bolster the cross-lingual capacities of these models, providing an equitable information ecosystem accessible to all.

a.Strengths:

S1: The research identifies and quantifies language disparities in LLM performance across multiple language pairs. This acknowledgment is crucial for understanding the limitations and challenges associated with linguistic diversity.

S2: This paper provides a universal multilingual LLM measurement framework, and the code is already open source, which provides a good contribution to multilingual LLM measurement in healthcare.

S3: This paper focuses on three essential criteria: correctness, verifiability, and consistency, providing a holistic approach to assessing LLMs in the healthcare domain.

b.Weaknesses:

W1: This paper lacks the compartment between XLingEval and other cross-language measurement frameworks.

W2: This paper is only limited to the medical field, but does not cover more medical content, so I am curious about whether the term 'medical' has meaning and whether it can be extended to a larger field.

**Questions:**

Q1:  To my knowledge, there are many cross language measurement frameworks such as XNLI, MLQA, XTREME, and XGLUE. What is the difference between XLingEval and them?

Q2: I am concerned about whether the language of the evaluated dataset will affect the evaluation results. There is a lack of good comparison in the evaluation, and there are flaws in the experimental design, which does not achieve strict comparison.

**Reviewer Confidence:**

3: The reviewer is confident but not certain that the evaluation is correct

**Scope:**

3: The work is somewhat relevant to the Web and to the track, and is of narrow interest to a sub-community

---

### Official Review · Reviewer_bdtt · 2023-11-23

**Novelty:** 7
**Technical Quality:** 7

**Review:**

This paper introduces XLingEval, a LLM evaluation framework designed to work in the healthcare domain.
Proposed framework is based on three main criteria, that is correctness, verifiability,
and consistency.
A further technical contribution of the paper is XLingHealth, a Cross-Lingual Healthcare benchmark for clinical health queries which has been evaluated in four main languages
Extensive experiments are provided.

I really liked the paper: it addresses an interesting, complex and topical research question. The title of the paper reflects well the research goals of the paper and, in my opinion, the paper advances the state of the art in the field of LLM in a specific and quite important domain, that is healthcare. I also greatly appreciated the presence of an anonymous GitHub, which allows other researchers to reproduce the experimental results obtained. The main limitation of the paper is related to its "proximity" to the conference topics: it is, in my opinion, a perfect paper for a Data Science, Machine Learning or NLP conference, but the adherence to the WebConf topics seems marginal to me. I also really appreciated the last part where the authors discuss the main limitations of their approach.

**Questions:**

Please better detail how the paper withs with the conference themes (and, in detail, with the Web)

**Reviewer Confidence:**

2: The reviewer is willing to defend the evaluation, but it is likely that the reviewer did not understand parts of the paper

**Scope:**

3: The work is somewhat relevant to the Web and to the track, and is of narrow interest to a sub-community

---

### Official Review · Reviewer_fv8C · 2023-11-23

**Novelty:** 6
**Technical Quality:** 6

**Review:**

The paper proposes a comprehensive evaluation framework called XLingEval for Large Language Models (LLMs) in the healthcare domain. The study highlights the importance of evaluating LLMs' responses in healthcare inquiries based on three fundamental criteria: correctness, consistency, and verifiability. The authors conducted experiments to compare LLMs' responses to expert-curated ground-truth answers across three healthcare datasets. The study also discusses the potential of LLMs in providing accessible healthcare information to the general public. However, the study also raises questions about equity in language technologies in healthcare, particularly for non-English languages. Overall, the paper provides insights into how well the general public can engage with and utilize LLMs in the healthcare domain.
The research questions addressed in this paper are:
1)	 How effective are LLMs as multilingual dialogue systems for healthcare queries?
2)	What are the fundamental criteria for evaluating LLM responses to naturalistic human-authored health-related questions?
3)	How do LLM responses compare across different languages and datasets?
4)	What are the implications of the findings for building equitable and accessible healthcare information systems?
The paper discusses three fundamental criteria for evaluating LLM responses in the context of healthcare queries. These criteria are: 1. Correctness: This pertains to the accuracy, comprehensiveness, and contextual appropriateness of LLMs' responses in healthcare inquiries. Ensuring correctness is essential due to the substantial implications associated with inaccuracies or errors in responses. 2. Consistency: This pertains to the coherence and stability of LLMs' responses across different healthcare inquiries. Ensuring consistency is important to build trust in LLMs and to ensure that users receive reliable information. 3. Verifiability: This pertains to the ability to provide evidence or sources to support LLMs' responses in healthcare inquiries. Ensuring verifiability is important to enable users to verify the accuracy and reliability of the information provided by LLMs.
It presents a framework for assessing the efficacy of LLMs in the healthcare domain and introduces criteria that can be modified to adapt to other critical domains such as legal, finance, and education. The paper also highlights the need for the adoption of cross-lingual analysis in frameworks to assess the capabilities and potential harms of LLMs.
This paper explains that investigating the effectiveness of LLMs as multilingual dialogue systems for healthcare queries is important for ensuring equity in the real-world use of these systems. The development of LLMs is disproportionately focused on English, and it remains unclear how these LLMs perform in the context of non-English languages. The study found a pronounced disparity in LLM responses across four major global languages, indicating a need for enhanced cross-lingual capabilities. Investigating the effectiveness of LLMs as multilingual dialogue systems for healthcare queries is important to ensure that LLMs can provide accurate and reliable information to people who may not have access to healthcare professionals or who may not be able to communicate effectively with healthcare professionals due to language barriers. It is also important to ensure that LLMs can provide equitable healthcare information accessible to all, regardless of language.

**Questions:**

How can the proposed framework,  XLINGEVAL, contribute to the actual improvement of the large language models?

**Reviewer Confidence:**

4: The reviewer is certain that the evaluation is correct and very familiar with the relevant literature

**Scope:**

4: The work is relevant to the Web and to the track, and is of broad interest to the community

---

### Official Review · Reviewer_HFyh · 2023-11-24

**Novelty:** 5
**Technical Quality:** 6

**Review:**

This paper introduces XLingEval, a novel framework for evaluating LLMs in the medical domain in a cross-lingual setting. XLingEval aims to quantify three desired criteria that LLMs should meet in order to be considered safe for medical applications: correctness, consistency, and verifiability. By proposing numeric metrics for all three of these criteria, the authors are able to for the first time quantify the performance difference that LLMs suffer when operating in languages other than English.

Pros:
- The paper provides a quantitative metric for studying the problem of LLM safety and reliability, which is widely regarded as an important problem but thus far has largely been examined qualitatively. This could be of broad interest not only to researchers in the medical domain but also other domains where issues like correctness are of importance in evaluating the use of LLMs
- The paper's focus on direct comparison of performance across languages, while not completely novel, is still relatively unique, and the results add some valuable quantitative evidence for claims that LLMs suffer performance hits in cross-lingual settings.

Cons:
- Human validation is included only for the "correctness" setting. I feel that human validation would also be valuable for the "consistency" setting, which currently relies on instruments such as topic models and embeddings that, while widely used and known to be fairly reliable, are also known to make mistakes and not always align with human notions of consistency.
- The decision to focus on correctness, consistency, and verifiability make intuitive sense but are not directly motivated by prior work. Given how much recent work there has been on LLM safety, I feel like the paper would benefit from expanding the prior work section to include citations that point to the importance of the three selected criteria, thus making it more clear how this paper fits into the existing work.

Overall, I feel that this paper's core contribution, XLingEval, is well-designed and would be of broad interest to the applied NLP community. Some improvements could be made to better situate the paper in the broader sphere of research on LLM safety, but this is a relatively minor issue and could reasonably be addressed within minor revisions after acceptance.

**Questions:**

Related to the last point listed above, I wonder how what your process was of selecting correctness, consistency, and verifiability as your three key metrics. Was this motivated by looking at other similar studies or prior work in this area? Or do you regard this selection of specific criteria as also being a novel contribution of this work?

**Reviewer Confidence:**

3: The reviewer is confident but not certain that the evaluation is correct

**Scope:**

4: The work is relevant to the Web and to the track, and is of broad interest to the community

---

### Official Review · Reviewer_5p7V · 2023-11-29

**Novelty:** 6
**Technical Quality:** 6

**Review:**

Quality: The paper is of high quality, as it provides a rigorous and comprehensive framework to evaluate the performance of large language models (LLMs) for healthcare queries in different languages. The paper uses three criteria to assess the LLMs: correctness, consistency, and verifiability. The paper also presents a novel cross-lingual benchmark for healthcare queries, called XLingHealth, which covers four languages: English, Spanish, Chinese, and Hindi. The paper reports the results of applying the framework to two LLMs, GPT-3.5 and MedAlpaca, and analyzes the strengths and weaknesses of each model. The paper follows the standard structure of a research paper, with a clear introduction, literature review, methodology, results, discussion, and conclusion. The paper also provides sufficient details and references to support its claims and arguments.
Clarity: The paper is well-written and easy to follow. The paper uses clear and concise language, and avoids unnecessary jargon and technical terms. The paper explains the motivation and objectives of the study, the main contributions and findings, and the implications and limitations of the work. The paper also uses appropriate figures, tables, and equations to illustrate and summarize the data and results. The paper follows the formatting and citation guidelines of the venue, and uses consistent notation and terminology throughout the paper.
Originality: The paper is original and innovative, as it addresses a timely and important problem of evaluating LLMs for healthcare queries in different languages. The paper proposes a new framework that considers not only the accuracy, but also the reliability and transparency of the LLMs. The paper also introduces a new cross-lingual benchmark for healthcare queries, which is the first of its kind and covers a diverse set of languages and topics. The paper demonstrates the applicability and usefulness of the framework and the benchmark, and reveals the current state and challenges of LLMs for healthcare queries in different languages.
Significance: The paper is significant and impactful, as it contributes to the advancement of LLMs for healthcare queries in different languages. The paper provides a valuable tool and resource for researchers and practitioners who want to evaluate and improve LLMs for healthcare queries in different languages. The paper also raises awareness and discussion on the safety and equity issues of LLMs for healthcare queries in different languages, and suggests directions for future work and improvement.
Pros and cons: The paper has the following pros and cons:
Pros:
- The paper proposes a novel and comprehensive framework to evaluate LLMs for healthcare queries in different languages, which considers correctness, consistency, and verifiability as the key criteria.
- The paper presents a new cross-lingual benchmark for healthcare queries, called XLingHealth, which covers four languages: English, Spanish, Chinese, and Hindi, and includes a variety of topics and query types.
- The paper reports the results of applying the framework to two LLMs, GPT-3.5 and MedAlpaca, and provides a detailed analysis and comparison of their performance and behavior across the evaluation metrics and languages.
- The paper highlights the need and potential to improve the cross-lingual capabilities of LLMs and to ensure equitable information access for all.
Cons:
- The paper focuses on healthcare queries, which is a specific and narrow domain, and may not generalize to other domains and tasks.
- The paper uses only two LLMs, GPT-3.5 and MedAlpaca, which are both based on the same architecture and pre-training data, and may not represent the diversity and variety of LLMs available and possible.
- The paper does not provide any user feedback or evaluation on the usefulness and satisfaction of the LLMs for healthcare queries in different languages, which is an important aspect of the user experience and acceptance of LLMs.

**Questions:**

- How do you ensure the quality and reliability of the human annotations for the XLingHealth benchmark? What are the qualifications and backgrounds of the annotators, and how did you measure their agreement and consistency?
- How do you handle the cases where the LLMs generate answers that are partially correct or partially contradictory to the ground-truth answers? How do you define the similarity and comprehensiveness of the answers in a rigorous and objective way?
- How do you account for the potential biases or errors in the translation models that you used to create the multilingual versions of the datasets? How do you ensure that the translation models do not introduce any noise or distortion in the meaning or fluency of the questions and answers?
- How do you compare the performance of LLMs across different languages and datasets? Do you use any cross-lingual evaluation metrics or baselines to measure the relative strengths and weaknesses of the LLMs in different languages and domains?
- How do you interpret and explain the results of the verifiability criterion? What are the factors that influence the LLMs’ ability to authenticate the validity of the claims? How do you measure the confidence or uncertainty of the LLMs in their predictions?

**Reviewer Confidence:**

3: The reviewer is confident but not certain that the evaluation is correct

**Scope:**

3: The work is somewhat relevant to the Web and to the track, and is of narrow interest to a sub-community

---

### Decision · Program_Chairs · 2024-01-22

**Decision:**

Accept (Oral)

**Comment:**

The reviewers mostly agree that this paper makes a strong contribution and is well executed.